# Phylogenomic analysis supports the ancestral presence of LPS-outer membranes in the Firmicutes

**Luisa CS Antunes[1†], Daniel Poppleton[1†], Andreas Klingl[2], Alexis Criscuolo[3], Bruno Dupuy[4], Céline Brochier-Armanet[5], Christophe Beloin[6], Simonetta Gribaldo[1*]**

[1]Unité de Biologie Moléculaire du Gène chez les Extrêmophiles, Département de Microbiologie, Institut Pasteur, Paris, France; [2]Plant Development and Electron Microscopy, Department of Biology I, Biocenter LMU, Munich, Germany; [3]Bioinformatics and Biostatistics Hub, Institut Pasteur, Paris, France; [4]Laboratoire Pathogenèse des Bactéries Anaérobies, Département de Microbiologie, Institut Pasteur, Paris; [5]Laboratoire de Biométrie et Biologie Évolutive, University Lyon I, Villeurbanne, France; [6]Unité de Génétique des Biofilms, Département de Microbiologie, Institut Pasteur, Paris, France

**Abstract** One of the major unanswered questions in evolutionary biology is when and how the transition between diderm (two membranes) and monoderm (one membrane) cell envelopes occurred in Bacteria. The Negativicutes and the Halanaerobiales belong to the classically monoderm Firmicutes, but possess outer membranes with lipopolysaccharide (LPS-OM). Here, we show that they form two phylogenetically distinct lineages, each close to different monoderm relatives. In contrast, their core LPS biosynthesis enzymes were inherited vertically, as in the majority of bacterial phyla. Finally, annotation of key OM systems in the Halanaerobiales and the Negativicutes shows a puzzling combination of monoderm and diderm features. Together, these results support the hypothesis that the LPS-OMs of Negativicutes and Halanaerobiales are remnants of an ancient diderm cell envelope that was present in the ancestor of the Firmicutes, and that the monoderm phenotype in this phylum is a derived character that arose multiple times independently through OM loss.

**\*For correspondence:** simonetta.gribaldo@pasteur.fr

[†]These authors contributed equally to this work

**Competing interests:** The authors declare that no competing interests exist.

## Introduction

The bacterial envelope is one of the oldest and most essential cellular components, involved in key housekeeping functions such as physical integrity, cell division, motility, substrate uptake and secretion, and cell-cell communication (*Silhavy et al., 2010*). Yet, bacteria show substantial differences in their cell envelope architectures, among which the most dramatic one is the presence of one (monoderm) or two (diderm) membranes (*Sutcliffe, 2010*). The study of cell envelope architecture has been mostly narrowed to the Firmicutes and the Gammaproteobacteria as textbook examples of monoderm and diderm bacteria, respectively. In *Bacillus subtilis*, teichoic and lipoteichoic acids are embedded in a thick peptidoglycan wall, while in *Escherichia coli* a thin peptidoglycan layer is surrounded by an outer membrane (OM) whose biogenesis and functioning involve a complex system of synthesis and transport for LPS, lipoproteins, and OM proteins (OMPs) (*Silhavy et al., 2010*).

The transition between monoderm and diderm cell envelopes must have been a significant and complex process in the evolutionary history of Bacteria. Two major hypotheses have been largely discussed in the literature, which can be generally defined as *diderm-first* (*Cavalier-Smith, 2006*)

**eLife digest** The cell envelope is one of the evolutionarily oldest parts of a bacterium. This structure – made up of a cell wall and either one or two cell membranes – surrounds the bacterial cell, maintaining the cell's structure and providing an interface through which bacteria can sense their environment and communicate.

Bacteria can be broadly classed based on the number of cell membranes that their envelope consists of. Bacteria that have a single cell membrane are known as "monoderm", whereas those with two membranes are termed "diderm". The number of membranes that bacteria have can affect how well they resist antibacterial compounds. When, how and why bacteria switched between monoderm and diderm cell envelopes are some of the major unanswered questions in evolutionary biology.

The textbook example of a monoderm cell envelope can be found in bacteria called Firmicutes. This group includes some notoriously harmful bacteria such as *Staphylococcus*, which can cause conditions ranging from abscesses to pneumonia. However, some Firmicutes possess two cell membranes. It was unclear how these unusual diderm Firmicutes developed a second membrane, and how they are related to their monoderm relatives.

Antunes, Poppleton et al. set out to answer these questions by analyzing the information contained in the thousands of bacterial genomes that have already been described. The results indicate that Firmicutes originally had diderm envelopes, and that species with monoderm envelopes arose independently several times through the loss of their outermost membrane.

Future work is needed to investigate the driving forces and the precise mechanism that led most Firmicutes to lose their outer membrane. Also, further characterization of diderm Firmicutes will provide key information about the biology of these poorly understood bacteria.

and *monoderm-first* (*Gupta, 2011*; *Lake, 2009*) scenarios. The fact that the majority of phyla seem to possess two membranes might favor the *diderm-first* scenario, although the actual diversity of cell envelopes in Bacteria remains largely unexplored (*Sutcliffe, 2010*). However, the lack of a robustly resolved phylogeny for Bacteria, notably the uncertainty on its root and the nature of the earliest branches, has left the relationships between diderm and monoderm phyla unclear, and not allowed to define in which direction and how many times this transition occurred.

In this respect, the Negativicutes (*Marchandin et al., 2010*) represent an interesting case: while belonging phylogenetically to the classical monoderm Firmicutes, they surprisingly display a diderm cell envelope with an OM and LPS (*Delwiche et al., 1985*; *Vos et al., 2009*). The Negativicutes have been identified in various anaerobic environments, such as soil and lake sediments, industrial waste, and animal digestive tract (*Vos et al., 2009*). Their best-characterized member is *Veillonella*, first described in 1898 by Veillon and Zuber (*Veillon and Zuber, 1898*). Curiously, the very first observation and use of the term 'outer membrane' has been based on studies of *Veillonella* (*Bladen and Mergenhagen, 1964*). *Veillonella* is one of the most abundant components of the human oral flora (*Tanner et al., 2011*), and a common inhabitant of the intestinal microbiome (*van den Bogert et al., 2013*). Together with other gut microbes, it has been recently associated with maturation of the immune system and partial protection of asthma in infants (*Arrieta et al., 2015*), but can also develop into an opportunistic pathogen (*Hirai et al., 2016*). Several other Negativicutes members such as *Dialister, Selenomonas, Mitsuokella,* and *Anaeroglobus* show increased incidence in oral tract disease linked to biofilm formation (*Griffen et al., 2012*) and involvement in other infections (*Wang et al., 2015*). Very little experimental data is available on the nature of the diderm cell envelope of Negativicutes. In *Selenomonas ruminantium* the abundant OmpM protein appears to replace the important function of Braun's lipoprotein in anchoring the OM to the cell peptidoglycan through a link with cadaverine (*Kojima et al., 2010*).

How the OM originated in the Negativicutes represents an evolutionary conundrum. Recently, Tocheva and colleagues analyzed the sporulation process in the Negativicute *Acetonema longum* by cryoelectron microscopy (*Tocheva et al., 2011*). They showed that, while an outer membrane forms only transiently during sporulation in classically monoderm Firmicutes such as *Bacillus subtilis*, it is

retained in *A. longum* leading to its diderm phenotype (*Tocheva et al., 2011*). This study provided the first experimental support for the hypothesis that the bacterial OM could have initially evolved in an ancient sporulating monoderm bacterium (*Dawes et al., 1980*; *Errington, 2013*; *Vollmer, 2012*). Moreover, a phylogenetic tree of the essential Omp85 protein family for proteins insertion in the outer membrane, although largely unresolved, did not show the Negativicutes as emerging from any specific diderm phylum (*Tocheva et al., 2011*). The authors speculated that the OM of Negativicutes was not acquired by horizontal gene transfer but was already present in the ancestor of Firmicutes and would have been lost in the other members of this phylum, although it remained unclear when and how many times this would have occurred (*Tocheva et al., 2011*). In contrast, a recent analysis of the genome of the Negativicute *Acidaminococcus intestini* revealed that as much as 7% of the BLAST top hits were from Proteobacteria, the majority of which corresponded to functions related to OM biogenesis, concluding to a possible acquisition of the OM in Negativicutes by horizontal gene transfer (*Campbell et al., 2014*).

Interestingly, the Negativicutes are not the only diderm lineage in the Firmicutes. The Halanaerobiales are a poorly studied group of moderate halophilic, strictly anaerobic Firmicutes that were isolated from saline environments such as lake and lagoon sediments, and oil reservoirs (*Oren, 2006*; *Roush et al., 2014*). Similarly to the Negativicutes, they display a diderm-type cell envelope, with a thin peptidoglycan and an outer membrane (*Cayol et al., 1994*; *Zeikus et al., 1983*; *Zhilina et al., 1992*, *Zhilina et al., 2012*). When analyzing the first sequenced genome of a member of Halanaerobiales, *Halothermothrix orenii*, Mavromatis and colleagues identified a number of OM markers, suggesting the presence of an LPS-diderm cell envelope homologous to the one of Negativicutes and other diderm bacteria (*Mavromatis et al., 2009*). In contrast to the few analyses on Negativicutes, no experimental data are available on the characteristics of the OM in the Halanaerobiales.

The existence of two diderm lineages in the Firmicutes provides a fantastic opportunity to clarify the monoderm/diderm transition in this major bacterial phylum. However, the origins and evolutionary relationships between the OM of Halanaerobiales and Negativicutes have been unclear. In fact, no Halanaerobiales were present in the analysis of Tocheva (*Tocheva et al., 2011*). Mavromatis et al. built a tree from the combined analysis of the genes coding for LPS, which showed a clustering of Halanaerobiales and Negativicutes, leading the authors to propose a horizontal gene transfer of the OM between these two lineages (*Mavromatis et al., 2009*). However, the sequenced genome of only one member of Halanaerobiales and one of Negativicutes were available at the time, and the LPS tree was largely unresolved (*Mavromatis et al., 2009*). Moreover, current phylogenies of the Firmicutes have been unclear with respect to the relationships between Negativicutes and Halanaerobiales. The Negativicutes have been alternatively indicated as branching within Clostridia (*Yutin and Galperin, 2013*; *Mavromatis et al., 2009*; *Vesth et al., 2013*) or at the base of Bacilli (*Kunisawa, 2015*). The phylogenetic placement of Halanaerobiales remains also uncertain, as they have been assigned either to Class Clostridia (*Cayol et al., 1994*), as a deep branch in the Firmicutes (*Mavromatis et al., 2009*; *Vos et al., 2009*; *Kunisawa, 2015*), or left unresolved (*Yutin and Galperin, 2013*). Finally, no detailed genomic analysis has been carried out to infer and compare the characteristics of the cell envelopes of several Halanaerobiales and Negativicutes.

The large number of Negativicutes and Halanaerobiales genomes currently available prompted us to carry out a global phylogenomic study. This allowed to robustly clarifying the relative placement of Negativicutes and Halanaerobiales within the Firmicutes, to assess the evolutionary relationships of their cell envelopes, and to perform in depth comparative analysis to understand the characteristics of key OM-related processes in these two lineages. Our results provide robust support for an emergence of monoderm Firmicutes from diderm ancestors via multiple independent losses of the OM.

## Results

### Electron microscopy of the diderm cell envelopes of Halanaerobiales and Negativicutes

Although the presence of an OM has been previously shown by electron microscopy for members of Negativicutes (e.g. *Tocheva et al., 2011*) and Halanaerobiales (e.g. *Zhilina et al., 2012*), these images have been obtained separately and with different techniques, making difficult their

comparison. We therefore obtained electron microscopy images of one representative of Negativicutes (*Megamonas rupellensis*) and one of Halanaerobiales (*Halanaerobium saccharolyticum*). We used transmission electron microscopy (TEM) following high-pressure freezing, freeze substitution, plastic embedding and ultrathin sectioning of the samples (see Materials and Methods). The application of high-pressure freezing in combination with appropriate freeze-substitution protocols facilitates the ultrastructural analysis of microorganisms and their membranes and also results in densely and homogeneously packed cytoplasm (*McDonald, 2007*; *McDonald et al., 2007a*; *Rachel et al., 2010*).

Ultrathin sections of high-pressure frozen cells of *M. rupellensis* and *H. saccharolyticum* confirmed the presence of clearly diderm-type cell envelope architecture in both strains (*Figure 1*). In a cross section from inside to outside, the densely packed cytoplasm of *M. rupellensis* is surrounded by a cytoplasmic membrane followed by the periplasm with a thin peptidoglycan layer and an OM (*Figure 1A*). Furthermore, pilus-like structures could be detected, as well as another electron dense layer outside the OM, which might correspond either to the lipopolysaccharide (LPS) or an S-layer. Thin sections of *H. saccharolyticum* also revealed a diderm cell envelope with a densely packed cytoplasm enclosed by a membrane surrounded by a relatively electron lucent periplasm so that the thin line representing the peptidoglycan is clearly visible (*Figure 1B*). For both organisms, in some cases the periplasm appeared inflated (*Figure 1C and D*), which was most likely caused by a preparation artifact due to swelling of the cells in the freeze substitution process. This effect nevertheless enabled us to observe the peptidoglycan much better as compared to cells without that artifact.

## Robust phylum-level phylogeny of the Firmicutes supports distinct origins of Halanaerobiales and Negativicutes

We gathered homologues of 47 ribosomal proteins from a local database of 205 Firmicutes taxa and 13 bacteria belonging to eight major phyla as outgroup (Materials and methods). We did not include the Tenericutes in the analysis, because their reduced genomes and fast evolutionary rates are likely to cause artifacts in deep phylogenies, but it is known that they phylogenetically belong to the Bacilli (*Davis et al., 2013*). We assembled the 47 ribosomal proteins into a large concatenated dataset (5551 amino acid characters) and carried out Bayesian analysis with a sophisticated site-heterogeneous model of protein evolution (CAT) that allows each site to evolve under its own substitution matrix and is robust against tree reconstruction artifacts that frequently affect deep phylogenies (*Lartillot and Philippe, 2004*). The Bayesian tree was well resolved at most nodes (Posterior Probabilities (PP) > 0.95, *Figure 2*). Despite the weak signal and the stochastic errors frequently associated to small proteins such as ribosomal ones, topology congruence tests on individual markers showed a largely congruent phylogenetic signal, especially at high rank taxonomy level, justifying their combined analysis (*Figure 2—figure supplement 1* and Materials and methods). Maximum Likelihood (ML) analysis of the same concatenated dataset and the site-homogeneous LG model (*Le and Gascuel, 2008*) gave a largely consistent topology although it was much less resolved, especially at deep nodes (*Figure 2—figure supplement 2*). With respect to previous analyses, the relative placement of Negativicutes and Halanaerobiales in the Firmicutes phylogeny was robustly resolved (*Figure 2*). In fact, the Negativicutes branched within Class Clostridia, specifically related to Peptococcaceae and other incertae-sedis clostridial families (PP = 1, *Figure 2*). This placement is consistent with previous analyses, although performed with less taxa (*Yutin and Galperin, 2013*; *Mavromatis et al., 2009*). As opposed to the diderm nature of Negativicutes, the members of Peptococcaceae have monoderm phenotype (*Vos et al., 2009*) and no homologues of OM markers.

In contrast, the Halanaerobiales emerged as a distinct, well-supported, and deep-branching lineage of the Firmicutes (PP = 0.99, *Figure 2*), robustly grouped with the order Natranaerobiales (PP = 1, *Figure 2*). This clustering was also observed in a previous analysis performed with only one member of Halanaerobiales and one of Natranaerobiales, and its position in the Firmicutes phylogeny was left unresolved (*Yutin and Galperin, 2013*). Natranaerobiales are a poorly known group of moderately halophilic Firmicutes that appear monoderm under the microscope (*Mesbah et al., 2007*) and have no homologues of OM markers.

In order to verify the robustness of the distinct branching of the two diderm Firmicutes lineages, we ran AU tests on 12 topologies alternative to the Bayesian ribosomal protein concatenate tree, where Negativicutes or Halanaerobiales were moved 'up and down' the six nodes separating them ($N_1$-$N_6$ for the topologies involving moving the Negativicutes; $H_1$-$H_6$ for the topologies

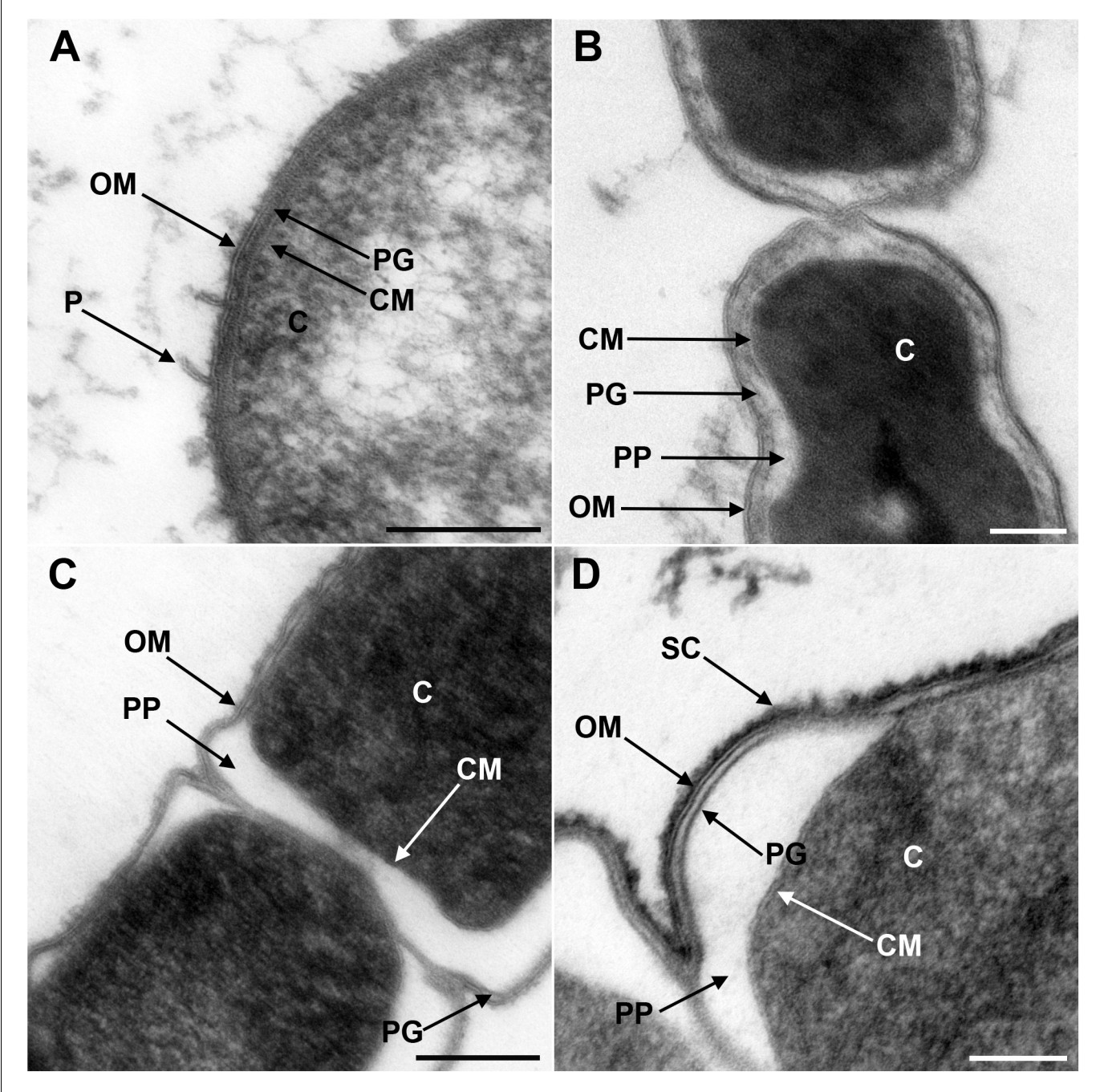

**Figure 1.** Transmission electron microscopy of a member of Negativicutes and a member of Halanaerobiales. Ultrathin sections of high-pressure frozen cells of the Negativicutes member *Megamonas rupellensis* (**A,C**), and the Halanerobiales member *Halanaerobium saccharolyticum* (**B,D**). A Gram-negative like cell wall architecture is visible for both taxa (**A,B**): a cytoplasmic membrane (CM) surrounding the cytoplasm (C), a thin peptidoglycan layer (PG), and an outer membrane (OM). Pili-like structures (P) are also visible in *M. rupelllensis*. In some cases and due to a preparation artifact caused by swelling of the cells, the OM detaches from the IM creating an enlarged periplasmic space (PP) between two dividing cells (**C,D**). In these cases, the peptidoglycan becomes more apparent as it is also the case for an electron dense surface coat (SC), which might represent lipopolysaccharide (LPS) or a potential S-layer. Scale bars: 200 nm (**A,C**) and 100 nm (**B,D**).

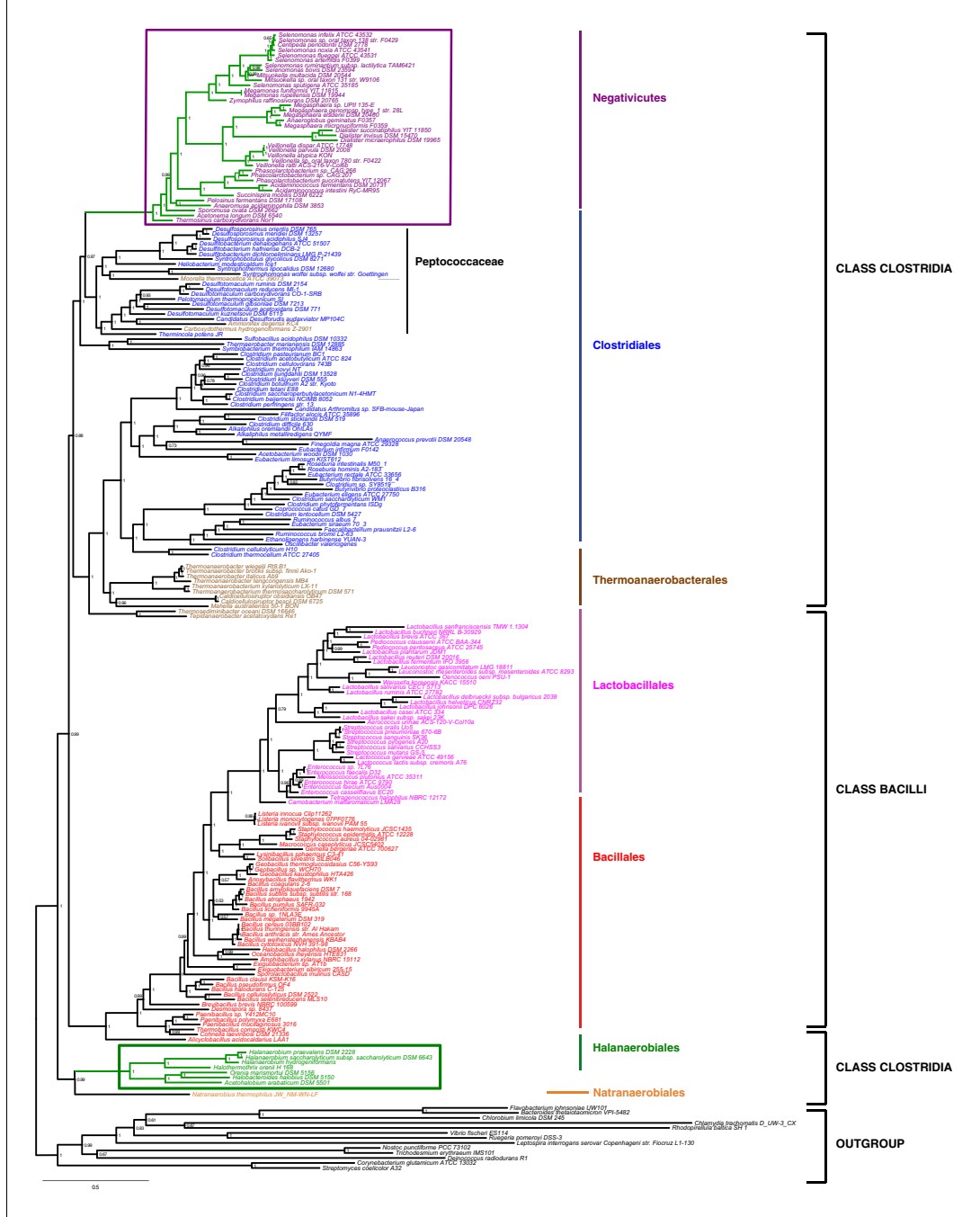

**Figure 2.** Phylum-level phylogeny of the Firmicutes. Bayesian phylogeny of the Firmicutes based on a concatenation of 47 orthologous ribosomal proteins comprising 5551 amino acid positions and the CAT+GTR+Γ4 model. Values at nodes represent Bayesian posterior probabilities. The scale bar represents the average number of substitutions per site. For details on analyses, see Materials and methods.

The following figure supplements are available for figure 2:

**Figure supplement 1.** Results of IC congruence test for the 47 ribosomal proteins.

**Figure supplement 2.** Maximum likelihood phylogeny of the Firmicutes.

**Figure supplement 3.** Results of AU test for 12 alternative topologies.

involving moving the Halanaerobiales *Figure 2—figure supplement 3*, Materials and methods and Additional Data). Unfortunately, topology testing is currently only available in a Maximum Likelihood framework with site-homogeneous models. Accordingly, these tests should reflect the poor resolution of the deep nodes of the maximum likelihood tree (*Figure 2—figure supplement 2*). Nevertheless, the Bayesian topology of *Figure 2*($H_0N_0$) was the preferred one (*Figure 2—figure supplement 3*). Two alternative topologies only ($N_1$ and $N_2$) were not rejected by the data, where the Negativicutes branched earlier in the Clostridia. Importantly, the two alternative topologies presenting a clustering of Halanaerobiales and Negativicutes ($H_6$ and $N_6$) were strongly rejected by the data, as well as all topologies where the Halanaerobiales were moved away from the root of the Firmicutes tree ($H_2$-$H_5$) (*Figure 2—figure supplement 3* and Additional Data), consistent with the separate origins of the two diderm lineages.

To sum up, our phylogenetic analysis shows that Halanaerobiales and Negativicutes have distinct evolutionary origins, and are each related to different monoderm Firmicutes lineages.

## The LPS-OM of Negativicutes and Halanaerobiales are homologous structures with an ancient origin

In contrast to the distinct emergence of Halanaerobiales and Negativicutes in the Firmicutes, the presence in their genomes of markers related to OM biogenesis and functioning (*Campbell et al., 2014*; *Mavromatis et al., 2009*; *Tocheva et al., 2011*) clearly indicates that their diderm cell envelopes are homologous structures. However, as discussed in the Introduction section, the specific evolutionary relationships between the OMs of Halanaerobiales and Negativicutes have been unclear. Interestingly, synteny analyses revealed a large genomic locus that is conserved between Halanaerobiales and Negativicutes, and is not present in monoderm Firmicutes (*Figure 3* and *Supplementary file 1*). Other than LPS synthesis and transport (green), the genes belonging to this genomic locus encode a number of cell envelope systems, such as OMP assembly (blue), motility (light pink), OM-PG attachment (red), efflux (purple), but also a number of hypothetical proteins (brown), and proteins not known to be specifically related to the OM (white).

Such clustering is unusual as in *E. coli* for example these genes are scattered in different regions of the genome. However, the genes coding for the first four steps of LPS synthesis (*lpxABCD*) display a conserved synteny in diderm Bacteria at very large evolutionary distances, suggesting that they have similar evolutionary histories (*Opiyo et al., 2010*). Accordingly, synteny is also conserved

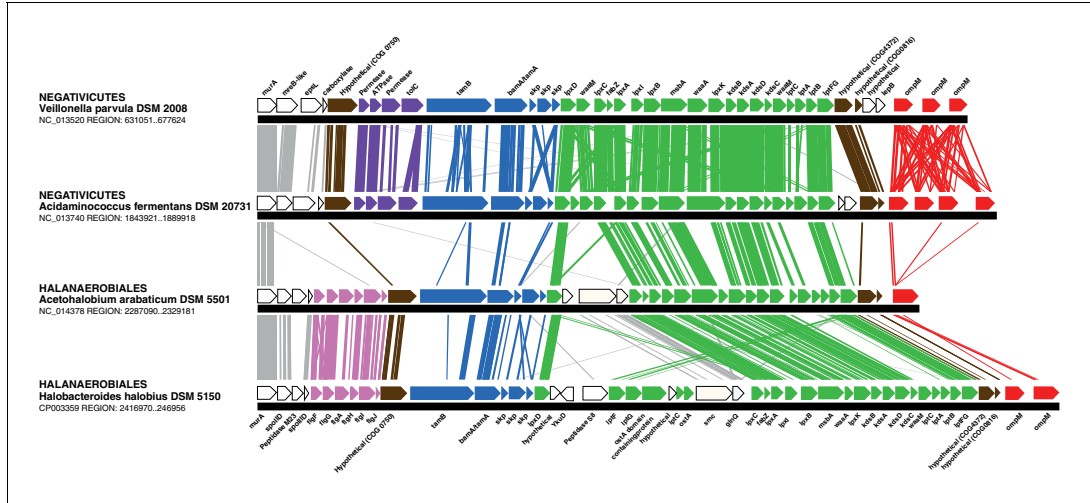

**Figure 3.** Conserved genomic locus for cell envelope components. Co-localization of the genes coding for LPS synthesis and transport, OMP assembly and structural OMPs in the Negativicutes and the Halanaerobiales. Representatives of the 2 families of Negativicutes and the 2 families of Halanaerobiales are shown (for full distribution and accession numbers see *Supplementary file 1*). Genes are colored according to their functional class: LPS synthesis and transport (green), OMP assembly (blue), flagellum (light pink), OM-PG attachment (red), hypothetical (brown), efflux (purple) (see text for discussion). White boxes indicate proteins not known to being related to the OM or non-conserved proteins whose connection with the OM is unclear. The figure was obtained by EasyFig (*Sullivan et al., 2011*), where vertical lines represent BLAST hits with a cutoff of 0.0001.

in Halanaerobiales and Negativicutes (*Figure 3* and *Supplementary file 1*). We therefore searched for these four core LPS genes (*lpxABCD*) in a local databank of 121 genomes representative of 30 major bacterial phyla (Materials and methods and *Supplementary file 2*). As compared to what could be previously inferred from available genomic data (*Sutcliffe, 2010*; *Opiyo et al., 2010*), we show the presence of homologues of the four core LPS coding genes in 26 major bacterial phyla, eight of which evidenced for the first time: Thermodesulfobacteria, Fibrobacteres, Ignavibacteria, Nitrospina, Chrysiogenetes, Cloacimonetes, Atribacteria, and Armatimonadetes (*Supplementary file 2*). This suggests that LPS-diderm cell envelopes might be even more widespread in Bacteria than currently thought, leaving only four major phyla that appear to lack the coding capacity for LPS: Thermotogae, Caldiserica, Chloroflexi/Thermomicrobia, and Actinobacteria (*Supplementary file 2*). We assembled the four core LPS protein homologues into a concatenated dataset (898 amino acid characters) also including Halanaerobiales and Negativicutes, and obtained a Bayesian tree with the CAT+GTR+$\Gamma_4$ evolutionary model (*Figure 4*, Materials and methods). In agreement with their conserved synteny, congruence tests showed that these four core LPS genes have a consistent phylogenetic signal at large evolutionary distances, in particular concerning the monophyly of major bacterial phyla, justifying their combined analysis (Materials and methods and *Figure 4—figure supplement 1*). Consistently with the notorious difficulty in resolving the global phylogeny of Bacteria, the tree is not completely resolved. However, it is largely in agreement with bacterial systematics, showing the monophyly of major phyla (*Figure 4*). This pattern indicates that the core LPS genes were present in the ancestor of each of these diderm phyla, and that inter-phylum horizontal gene transfers were surprisingly rare during bacterial evolution. Consistently, the Halanaerobiales and Negativicutes also form a well-supported monophyletic cluster (PP = 1, *Figure 4*), with internal branching pattern matching their respective reference species phylogeny shown in *Figure 2*.

These results indicate that the LPS-OM of Halanaerobiales and Negativicutes do not have distinct origins, but rather that, similarly to the other main diderm bacterial phyla, they were inherited from their common ancestor, which is also the ancestor of all Firmicutes, in agreement with *Tocheva et al., 2011*. The inclusion of a second diderm lineage in our analysis allows us to strengthen and extend this scenario, and to infer that present-day monoderm Firmicutes would have emerged from diderm ancestors via not less than five independent losses of the OM (*Figure 4B*). The two alternative topologies that were not rejected by AU tests do not affect the inference of a diderm ancestor and imply four and three independent OM losses, respectively (*Figure 2—figure supplement 3* and Additional Data).

## Outer membranes in a monoderm context

Our phylogenetic analyses suggest that the diderm cell envelopes of Halanaerobiales and Negativicutes might be the remnants of ancient bacterial structures that were inherited from the Firmicutes ancestor. In the absence of experimental characterization, exploration of genomic data can guide inferences on the nature of these atypical diderm cell envelopes. To this aim, we investigated a few key processes that are related to OM biogenesis and functioning and are shared between Negativicutes and Halanaerobiales. Because OM markers frequently display low sequence conservation or are part of large membrane-related protein families often preventing the building of robust phylogenies, we helped tentative annotation by merging information obtained from homology to known OM markers, the presence of specific protein domains, and genomic synteny. In this respect, the presence of the conserved OM locus helped annotation greatly and provided important insights into the unique nature of the cell envelopes of Negativicutes and Halanaerobiales, which show both specific characteristics as well as an intriguing combination of diderm and monoderm features (*Figure 5*).

### Diderm Firmicutes synthesize and transport LPS to the OM

LPS is a complex glycolipid exclusively present in the outer leaflet of the OM. Although LPS can be very heterogeneous in bacteria, it has an overall conserved structure composed of a membrane-anchored hydrophobic domain (lipid A, or endotoxin), an oligosaccharide (inner and outer core), which can be extended with a distal polysaccharide (O-antigen) (*Wang et al., 2010*). The Lipid A-core portion (known as 'rough' LPS) and the O-antigen have independent pathways for their

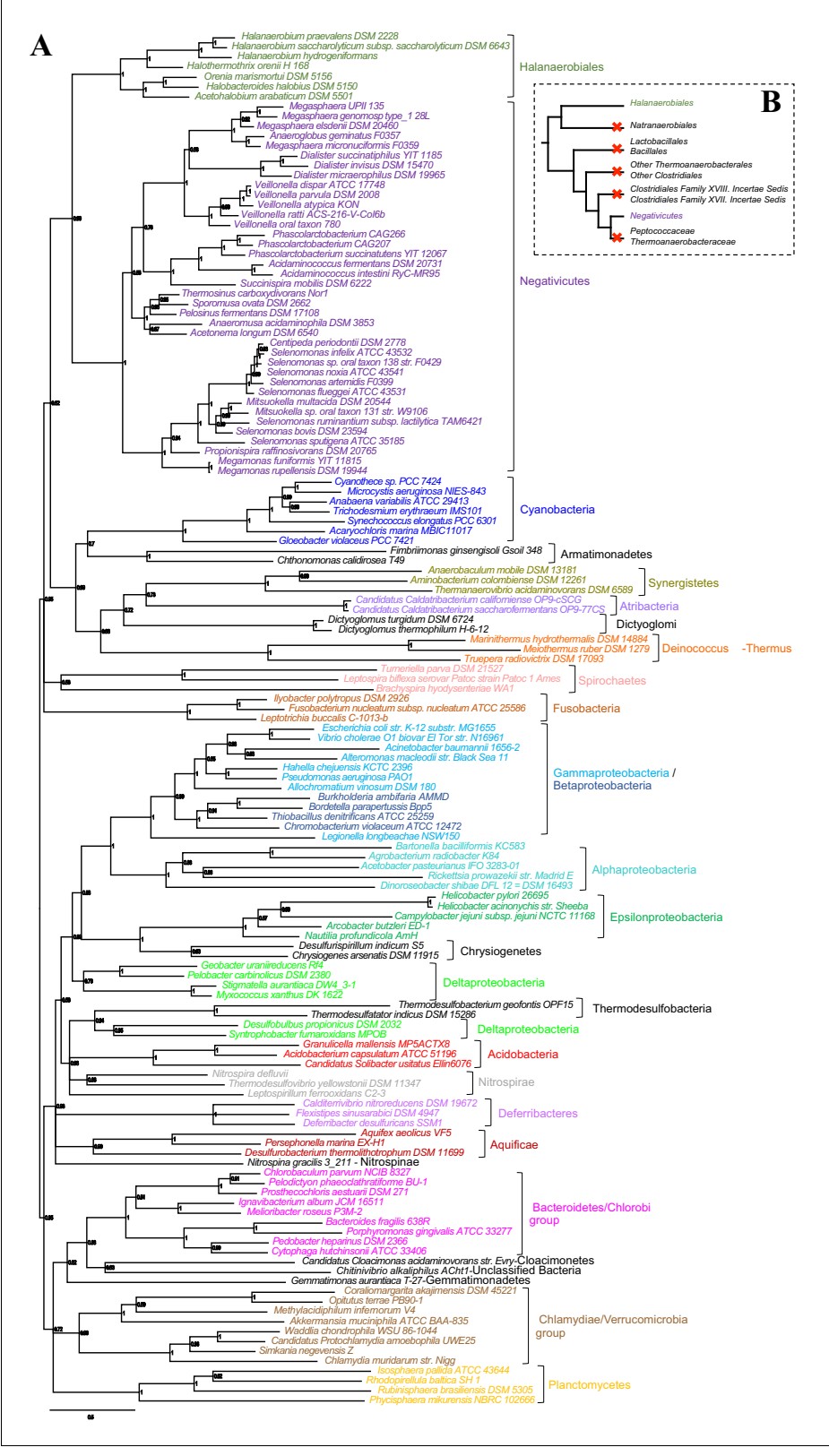

**Figure 4.** Phylogenetic tree of core LPS components. (**A**) Bayesian phylogeny based on a concatenation of orthologs of the four core components of the LPS biosynthesis pathway (*lpxABCD*), comprising 898 amino acid positions and the CAT+GTR+Γ4 model. Values at nodes represent Bayesian posterior probabilities. The scale bar represents the average number of substitutions per site. For details on analyses, see Materials and methods. (**B**)
*Figure 4 continued on next page*

*Figure 4 continued*

Schematic representation of the Firmicutes phylum-level phylogeny from **Figure 2**, onto which putative losses of the OM are mapped (red crosses). See text for discussion.

The following figure supplement is available for figure 4:

**Figure supplement 1.** Results of IC congruence test for the 4 LPS core proteins.

synthesis and transport. If O-antigen is produced, it is ligated to the lipid A-core by an integral

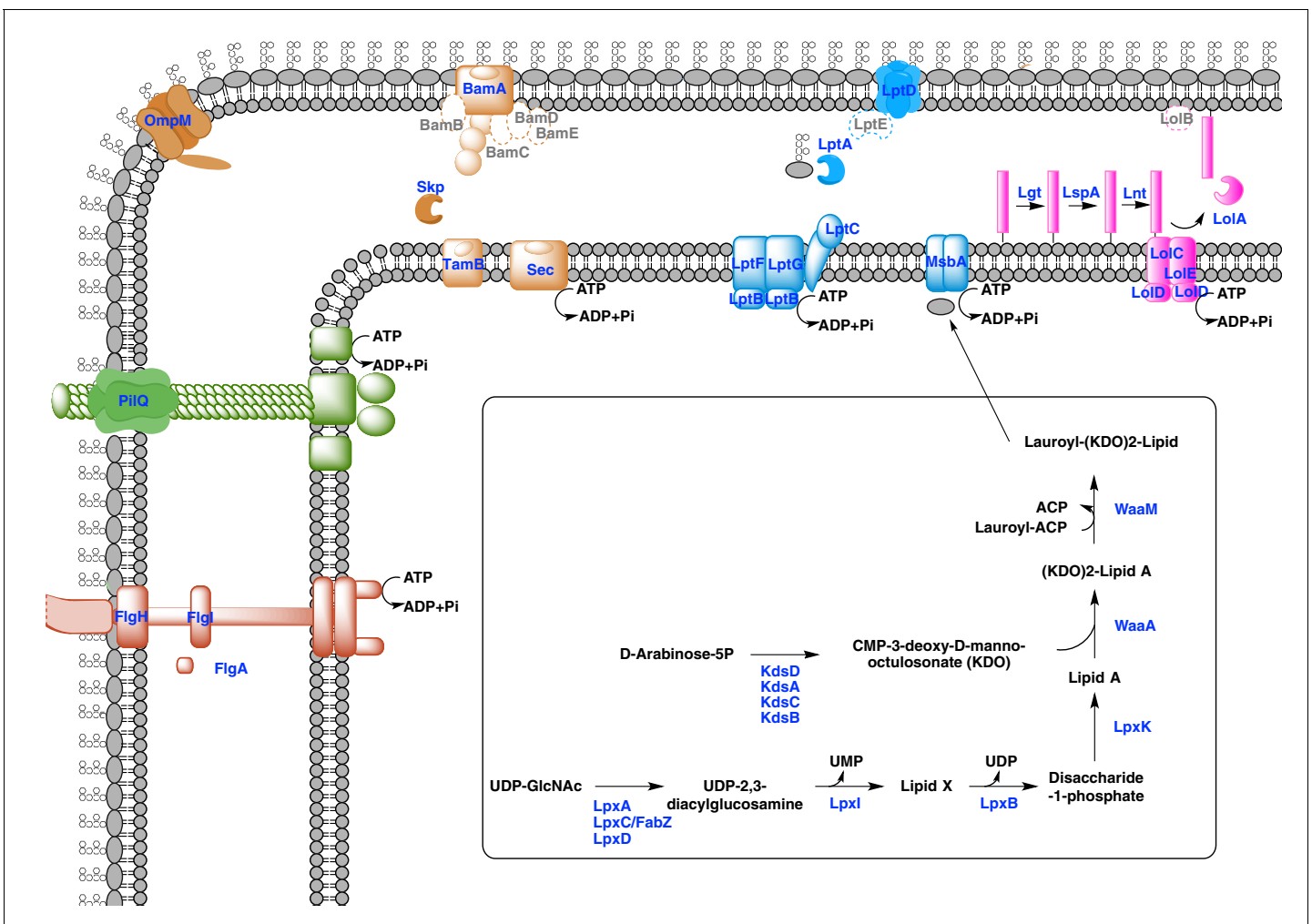

**Figure 5.** Sketched diagram of inferred characteristics of the diderm Firmicutes cell envelope. The main processes discussed in the text are shown schematically. Components that were not detected in the genomes of Negativicutes and Halanaerobiales are indicated with a dashed outline and grey font.

The following figure supplements are available for figure 5:

**Figure supplement 1.** Flagellar gene cluster of Negativicutes and Halanaerobiales.

**Figure supplement 2.** Genomic context of the genes coding for flagellar rings in Halanaerobiales and Negativicutes.

**Figure supplement 3.** Structure of the main Type IV pilus cluster in Negativicutes and Halanaerobiales.

membrane ligase, generating 'smooth' LPS. Then, the transport of LPS to the OM is carried out by a dedicated system (the Lpt pathway) and is independent from O-antigen presence (*Wang et al., 2010*).

Both Negativicutes and Halanaerobiales have a complete set of genes for the synthesis of the Lipid A moiety and inner core of LPS (*Figure 5*). These are clustered on their genomes (*lpxACDIBK/ waaM/waaA/kdsABCD*) (*Figure 3* and *Supplementary file 1*). The lack of *lpxH* and the presence of *lpxI* suggests that they use a recently-described alternative route for lipid A-core synthesis found in members of various bacterial lineages (*Metzger and Raetz, 2010*). Moreover, the *lpx* gene cluster also includes a *fabZ* homologue, which encodes an enzyme involved in a key step of fatty acid synthesis and is also present in monoderm Firmicutes (*Parsons and Rock, 2013*). Similarly to what is observed in *E. coli*, this *fabZ* homologue shows a conserved position on the chromosome next to *lpxC*, which catalyzes the first committed step in lipid A biosynthesis and is subject of tight regulation by the protease FtsH (*Führer et al., 2006*). Because FabZ acts on a substrate that is shared with the lipid A pathway, its conserved genomic proximity with LpxC is a strong indication that these two proteins interact in diderm Firmicutes, likely in a common process that regulates the phospholipid/ LPS balance of the OM as described in *E. coli* (*Klein et al., 2014*; *Ogura et al., 1999*).

Once synthesized, the lipidA-core is translocated through the IM by the flippase MsbA, which is composed of a typical architecture including an N-terminal ABC-transporter transmembrane domain and a C-terminal ATP-binding cassette domain (*Ruiz et al., 2009*). We found homologues with the same domain arrangement within the LPS synthesis gene cluster (*Figure 3* and *Supplementary file 1*). Although these are part of the very large protein family of ABC transporters that is also present in monoderm Firmicutes, their genomic location suggests that they are likely *bona fide* MsbA functional equivalents in diderm Firmicutes (*Figure 5*).

LPS is then matured and transported across the periplasm to the OM via the Lpt pathway (*Greenfield and Whitfield, 2012*; *Polissi and Sperandeo, 2014*; *Ruiz et al., 2009*). LPS is extracted from the IM by LptC and the LptFGB transporter, then mobilized to the OM by the chaperone LptA, and finally assembled into the OM by LptDE. In all Negativicutes and Hanaerobiales, we identified a conserved Lpt four-gene cluster next to the LPS synthesis genes (*Figure 3* and *Supplementary file 1*). This includes a homologue of LptB, a single homologue of LptF/G, plus two proteins of ~200aa containing OstA domains that may represent LptA and LptC (*Figure 5*). Apart from LptB, which belongs to the large P-loop-NTPase superfamily, none of the remaining putative components are present in monoderm Firmicutes. However, we could not find any clear homologues of the OM components LptD/E in the gene cluster or elsewhere in the genomes. Either these are too distantly related to being identified by sequence similarity, or Negativicutes and Halanaerobiales employ a non-homologous system to address LPS to the OM.

Concerning O-antigen, no obvious homologues of its synthetic pathway, as well as its transport through the IM and its ligation with the lipidA-core are present in the conserved gene cluster. Moreover, although we could identify some homologues in the genomes of Negativicutes and Halanaerobiales, these are part of very large protein families and are shared with other pathways, making it difficult to assess functional homology and infer with confidence if diderm Firmicutes are able to make smooth LPS.

## An ancient bi-functional Bam/Tam machinery?

The genomic locus also encodes homologues of two important components of the *E. coli* Sec-dependent OM protein assembly pathway (*Selkrig et al., 2014*): one copy of the major component of the Bam system (BamA), which is involved in protein translocation and OM assembly, and three copies of the periplasmic chaperone Skp (*Figure 3* and *Supplementary file 1*). However, we could not find any homologues of the associated lipoproteins of the Bam complex (BamBCDE) in the gene cluster nor elsewhere in the genomes of diderm Firmicutes, though it is known that these are not well conserved outside Proteobacteria (*Webb et al., 2012*).

In Proteobacteria, BamA is part of the large Omp85-family with function in protein translocation and OM assembly (*Heinz and Lithgow, 2014*; *Selkrig et al., 2014*). Members of this family have a conserved C-terminal OM-associated surface antigen domain, and polypeptide transport–associated (POTRA) domains of variable number that serve to interact with other proteins and accessory factors (*Webb et al., 2012*). In Proteobacteria, a close paralogue of BamA was recently discovered, called

TamA, which has one POTRA domain only, and was shown to be anchored to the OM and to form a two-pathway system together with the IM protein TamB to promote T5SS autotransporter assembly and secretion (*Selkrig et al., 2012*). Based on the large taxonomic distribution of TamB homologues, it has been recently proposed that this protein was present very early in bacterial evolution, and functioned with BamA, while TamA would have arisen more recently in Proteobacteria through gene duplication (*Heinz et al., 2015*).

We found that all analyzed Negativicutes and Halanaerobiales genomes possess a single BamA homologue, which is part of a gene cluster with a homologue of TamB, and up to three homologues of the Skp chaperone (*Figure 3* and *Supplementary file 1*). Such conserved synteny may indicate functional linkage. Interestingly, the same gene arrangement (TamB/BamA/Skp) has been observed in Spirochaetes (*Selkrig et al., 2012*), as well as in many other bacterial phyla (*Heinz et al., 2015*). The inference of a potential TamB/BamA/Skp system in the ancestor of the Firmicutes strengthens the hypothesis of an ancestral bi-functional role for BamA in both autotransporter secretion (together with TamB) and OMP assembly (together with Skp) (*Figure 5*).

## An ancestral system for peptidoglycan anchoring to the OM

The genomic locus also includes multiple homologues of OmpM (*Figure 3* and *Supplementary file 1*), a protein that appears to replace the function of Braun's lipoprotein in Negativicutes (see Introduction). In *S. ruminantium* OmpM results from the fusion of a C-terminal porin domain and an N-terminal S-layer homologous (SLH) domain with an unusual inward orientation towards the periplasm where it drives the correct anchoring of the OM to the peptidoglycan layer *via* specific interactions with the peptidoglycan-covalently-bound cadaverine (*Kalmokoff et al., 2009*; *Kojima et al., 2010*).

We found that Halanaerobiales also contain in the conserved genomic locus up to three proteins with an SLH domain (*Figure 3* and *Supplementary file 1*). These likely represent distant homologues of Negativicutes OmpM and are priority targets for experimental validation to confirm the presence of an OmpM-dependent system for OM anchoring in this deep-emerging lineage of the Firmicutes. Interestingly, there have been early reports of distantly related proteins with a domain arrangement similar to that of diderm Firmicutes OmpM and a proposed similar function in Cyanobacteria (*Hansel et al., 1998*) and in Thermotogae (*Engel et al., 1992*). OmpM-like proteins might therefore represent a widespread and ancient PG-OM anchoring system (*Figure 5*), possibly ancestral to the Braun's lipoprotein-based system.

## Monoderm appendages with diderm features

The monoderm-type flagellum has been mostly studied in *B. subtilis* (*Mukherjee and Kearns, 2014*). It is very similar in overall structure and number of components to the flagellum of diderm bacteria, to the exception of the absence of the rings P and L, the first spanning the periplasm and the second spanning the OM, which are coded by *flgH*, *flgI*, and *flgA* and are totally absent in monoderm flagella. Another difference is the absence of the rod cap and flagellar-specific peptidoglycan hydrolases, which allow insertion in the PG layer in diderm bacteria, and it is unknown how this process takes place in the thick PG layer of *B. subtilis* (*Mukherjee and Kearns, 2014*).

Most Negativicutes and Halanaerobiales strains analyzed in this study are flagellated (*Vos et al., 2009*) and have in fact a complete set of genes coding for the flagellum machinery, most of which embedded in an operon arrangement that is distinctive of *B. subtilis* and other monoderm Firmicutes, in particular the presence of the two rod assembly genes *flgB* and *flgC* (*Figure 5—figure supplement 1*) (*Mukherjee and Kearns, 2014*). This indicates that the flagella of diderm Firmicutes are closely related to those of their monoderm relatives. However, Negativicutes and Halanaerobiales also possess homologues of the three genes coding for the P and L rings typical of diderm flagella, *flgH*, *flgI*, and *flgA*, organized in a second conserved six-genes cluster that also contains the rod components *flgF*, *flgG*, and *flgJ* (*Figure 5—figure supplement 2* and *Supplementary file 1*). These data suggest that Negativicutes and Halanaerobiales might possess flagella with chimeric monoderm/diderm features (*Figure 5*). They may represent ancestral motility structures that adapted to loss of the OM in present-day monoderm Firmicutes lineages, possibly through a single excision of the *flgH*, *flgI*, and *flgA* gene cluster from the conserved genomic locus. Consistently with this hypothesis, we observed that representatives of the monoderm Firmicutes lineages that are most

closely related to Halanaerobiales (i.e. Natranaerobiales: *N. thermophilus*) and Negativicutes (i.e. Peptococcaceae: *Therminicola potens*) still retain an *flgJ* homologue (*Figure 5—figure supplement 1*) that may represent a remnant of the ancestral diderm system for rod insertion, which would have been lost in *B. subtilis* and other monoderm flagellated Firmicutes.

A similar, although less clear-cut, scenario could be inferred for type IV pili. These have very similar components in diderm and monoderm bacteria, to the exception of PilQ (also called secretin), which in diderm bacteria forms a channel that spans the OM and through which the pilus is assembled (*Melville and Craig, 2013*). In Firmicutes, it is unclear how the pilus passes through the thick peptidoglycan layer, and if a channel-forming protein is present (*Melville and Craig, 2013*). Both Negativicutes and Halanaerobiales have a conserved genomic locus that presumably codes for all essential components of a type IV pilus (*Figure 5—figure supplement 3*), consistently with the visualization of a potential pilus structure by microscopy in *Figure 1*. The gene cluster is similar to those present in their closely related monoderm relatives, to the exception of the inclusion of a homologue of PilQ, the secretin, which is present in all Negativicutes and Hanalerobiales (*Figure 5—figure supplement 3* and *Supplementary file 1*). These data suggest that the type IV pili of Negativicutes and Halanaerobiales may pass through the OM by a secretin-based mechanism similar to that of classical diderm bacteria (*Figure 5*).

## First suggestion of a Lol system in the Firmicutes

Lipoproteins are a major class of membrane proteins that play important physiological roles and are widespread in bacteria (*Buddelmeijer, 2015*; *Okuda and Tokuda, 2011*; *Sutcliffe et al., 2012*; *Zückert, 2014*). They are assembled via post-translational modifications after translocation through the inner membrane in both diderm and monoderm bacteria. The lipid modification occurs on a 'lipobox' signal near the N-terminus at the cytoplasmic membrane where the pre-lipoprotein diaglycerol transferase Lgt adds a diacylglycerol moiety. The lipobox signal is then cleaved off by the lipoprotein signal peptidase LspA, which allows anchoring of the lipoprotein to the outer face of the plasma membrane. A second lipoprotein N-acetyltransferase, Lnt, can add a third amide fatty acid onto the N-terminus of lipoproteins. Tri-acylated lipoproteins are preferentially targeted to the OM in diderm bacteria via the Lol machinery. This is a multiprotein system formed by an IM-spanning ABC-like transporter component (LolCDE) that captures lipoproteins from the IM and delivers them to the carrier LolA utilizing ATP hydrolysis. LolA then translocates the lipoprotein to the structurally related LolB, which inserts it into the OM by an unclear mechanism.

While LolCDE are part of the large protein family of ABC transporters, the presence of LolA homologues is indicative of a functional Lol system (*Sutcliffe et al., 2012*). We found no homologues of LolA nor of LolB in any of the Negativicutes genomes, confirming previous reports (*Campbell et al., 2014*). It is possible that Negativicutes do not address lipoproteins to the OM or that they use a still unknown, non-homologous system. In contrast, we found homologues of LolA in each of the analyzed Halanaerobiales genomes, which correlate to the presence of Lnt homologues (*Supplementary file 1*). By looking at the genomic surroundings of these LolA homologues, we identified a conserved gene cluster in most Halanaerobiales genomes, which likely code for a complete Lol system, notably the two ABC transporter permeases (LolC/E) and one ATPase (LolD) (*Supplementary file 1*). We could not find any homologues of LolB in the genomic locus or elsewhere in the Halanaerobiales genomes, although this protein is not well conserved in bacteria (*Sutcliffe et al., 2012*). These analyses suggest that Halanaerobiales may harbor a functional Lol system (*Figure 5*).

## New OM-related proteins?

Finally, three genes encoding proteins with unclear annotation display a conserved synteny within the large conserved genomic cluster (*Figure 3* and *Supplementary file 1*) and are present in all Negativicutes and Halanaerobiales genomes (*Supplementary file 1*), while absent from monoderm Firmicutes. The first one belongs to COG0816, which is annotated as 'Predicted endonuclease involved in recombination (possible Holliday junction resolvase in Mycoplasmas and *B. subtilis*)'. The second one belongs to COG4372, which is annotated as 'Uncharacterized conserved protein, contains DUF3084 domain'. The third one belongs to COG750, which is annotated as "Predicted membrane-associated Zn-dependent proteases 1", and is predicted to be in the OM by the PSORT

prediction software (Materials and methods). These proteins might be involved in OM biogenesis and functioning in diderm Firmicutes and are priority targets for experimental characterization.

## Discussion

The origin of the cell envelope has represented one of the most fascinating questions in evolutionary biology since decades, and has been widely discussed in the literature (*Blobel, 1980*; *Cavalier-Smith, 1987*, *2006*; *Errington, 2013*; *Griffiths, 2007*; *Gupta, 2011*; *Koch, 2003*; *Vollmer, 2012*). The main issue mostly revolves around the question of how and when an OM originated in Bacteria, and whether monoderm cell envelopes predate diderm cell envelopes or instead derived from them. The complexity of the diderm cell envelope with respect to a perceived more 'rudimentary' monoderm type, together with its higher resistance toward antibiotics, are arguments usually put forward to propose that the OM is a relatively late invention in Bacteria (*Koch, 2003*; *Gupta, 2011*). However, it is now evident that diderm phyla outnumber monoderm ones, and span a large fraction of bacterial diversity, including early emerging lineages (*Sutcliffe, 2010*; *Errington, 2013*). Unfortunately, the evolutionary relationships among monoderm and diderm bacterial phyla are presently unclear, and do not allow to clarify OM origins. In this respect, the existence of a major bacterial phylum –the Firmicutes- including both diderm and monoderm lineages, and whose evolutionary relationships can be resolved, provides a unique opportunity to address the issue.

Our results provide support for the hypothesis that the Firmicutes are ancestrally diderm, and that the monoderm envelope is a derived cell structure that originated through OM loss, at least in this phylum. Although previously suggested (*Tocheva et al., 2011*), the inclusion of both Halanaerobiales and Negativicutes in our analysis strengthens and extends this scenario. Our robust phylogeny of the Firmicutes indicates that Halanaerobiales and Negativicutes form two distinct lineages, each related to different monoderm relatives. This allows inferring that the OM was lost from three to five times independently in the Firmicutes, and is therefore not a unique event in the history of Bacteria that would have led to all present-day monoderm lineages, as proposed earlier (*Cavalier-Smith, 2006*). The deep branching of Halanaerobiales and the still limited genomic coverage for this group makes them a priority target for further exploration as their cell envelopes may retain ancestral characters.

Our results confirm that the LPS-OMs of Negativicutes and Halanaerobiales are homologous structures that share a common origin and are evolutionarily related with those of other classically diderm bacteria, therefore excluding convergence. Indeed, we show that the core enzymatic apparatus for producing LPS is even more widespread than previously thought, and that the LPS-OM is an ancient feature that emerged once and was largely inherited vertically during bacterial evolution, including in Halanaerobiales and Negativicutes. This is unusual for cytosolic enzymes, and underlines the key importance of maintaining cell-envelope function. Clearly, the availability of genomic data from an ever-wider sampling of bacterial diversity is sensibly changing perspective on the evolution of bacterial cell envelopes. For example, by revealing the presence of an LPS-OM in the ancestor of Deinococcus/Thermus we show that this phylum does not represent a monoderm-to-diderm intermediate (*Gupta, 2011*), but rather that LPS was lost in some of their members, a process similar to what likely occurred in Thermotogae (*Cavalier-Smith, 2006*). This indicates that, although having an OM is surely advantageous in certain conditions, the diderm cell envelope is a flexible structure that can be modified/simplified during evolution.

Although the presence of a large cluster coding for key OM functions might be seen as supporting the hypothesis of an acquisition of the OM of diderm Firmicutes via genetic transfer from a diderm bacterium (*Mavromatis et al., 2009*; *Campbell et al., 2014*), our data weaken this hypothesis. Given the distinct branching of the two diderm lineages in the Firmicutes phylogeny, and the pattern of LPS inheritance similar to all other diderm bacterial phyla, the gene transfer hypothesis would imply a complex scenario consisting of two independent transfers of a very large genomic region, a first one to the ancestor of Halanaerobiales or to the ancestor of Negativicutes, and a second one between these two ancestors, which would have had to coexist at the same time and in the same environment. The sudden acquisition of an OM has been already discussed as mechanistically complicated (*Cavalier-Smith, 2006*). This would have in fact required the dramatic modification of an originally monoderm cell envelope through the concerted acquisition of several complex systems at once, not to mention the replacement of the native inner membrane components of these

systems or their coordination with the newly acquired ones. Moreover, not all OM systems involved in OM biogenesis and function in Negativicutes and Halanaerobiales are part of the gene cluster, and their detailed annotation suggests that their cell envelopes share characteristics with deep-emerging bacterial phyla (e.g. the OmpM porin instead of Braun's lipoprotein for OM tethering), and present a peculiar combination of monoderm/diderm features (e.g. flagella, pili). The OM gene cluster may therefore represent an ancestral genomic locus for OM–related functions, and it becomes essential to obtain further experimental characterization of the OM of Halanaerobiales and Negativicutes, as well as many other poorly explored deep emerging bacterial phyla. Nevertheless, we have analyzed here only a few OM systems shared between Halanaerobiales and Negativicutes in order to infer the nature of the ancestral diderm cell envelope in the Firmicutes. There are surely additional components that are lineage-specific and may have been acquired from diderm bacteria thriving in the same environment. This is an important future area of investigation, as it could inform on how the presence of an OM in a Firmicutes background may have helped adaptation to specific niches, including the human environment.

By which process the ancestral OM would have been lost multiple times independently in the majority of present-day Firmicutes remains to be understood. It has been proposed that a spontaneous mutation leading to hypertrophy of the peptidoglycan layer would be sufficient to transform a diderm into a monoderm, through disruption of the attachment of the OM, leading to its loss (*Cavalier-Smith, 2006*). Alternatively, we speculate that mutations may have affected the ancestral OmpM, causing a de-regulation of OM-PG attachment. This transition may have been made easier during the process of sporulation, where an OM is transiently formed and lost when the vegetative cell matures. *Tocheva et al. (2011)* proposed indeed that the OM might have been lost in the Firmicutes to increase sporulation and germination efficiency (*Tocheva et al., 2011*). A link between OM loss and sporulation may explain why there is no current evidence of monoderm lineages within other diderm phyla that do not sporulate. Further genomic and experimental data on the closest monoderm relatives of Negativicutes and Halanaerobiales might provide key information on the process of OM loss.

Our results suggest that the cell envelopes of diderm Firmicutes might be the remnants of an ancient type of cellular structure, adding up to the ones found in the major diderm bacterial phyla. Moreover, they seem to have ancestral and simpler cell envelope systems with respect to the well-studied Proteobacteria. Halanaerobiales and Negativicutes are therefore promising new experimental models that will provide precious insights into the processes that have shaped the diversity not only of diderm cell envelopes, but also of monoderm ones.

## Materials and methods

### Ultrastructural analysis

Negativicutes strain *Megamonas rupellensis* DSM 19944[T] was grown anaerobically at 37°C to late exponential phase on TGY broth (w/v; 3% tryptone, 2% yeast extract, 0.5% glucose, 0.05% L-cysteine hydrochloride) as described previously (*Chevrot et al., 2008*). Halanaerobiales strain *Halanaerobium saccharolyticum subsp. saccharolyticum* DSM 6643[T] was grown anaerobically to late exponential phase at 37°C on a synthetic medium as described in (*Zhilina et al., 2012*). The ultrastructure of the respective bacterial strains was determined via transmission electron microscopy (TEM) following high-pressure freezing, freeze substitution, plastic embedding and ultrathin sectioning of the samples. Due to a relatively long transportation time to the high-pressure freezer, cells were pre-fixed with 2% glutaraldehyde. Afterwards, they were centrifuged for 10 min at 14.000 x g, the supernatant was discarded and the resulting pellet was resuspended in 50µl growth medium. From this cell suspension, 2µl were high-pressure frozen and freeze substituted as described in (*Peschke et al., 2013*). For substitution, acetone containing 0.2% $OsO_4$, 0.25% uranyl acetate and 5% (vol/vol) $H_2O$ was used. Embedding of the cells, sectioning and post-staining was carried out as described in (*Rachel et al., 2010*). For negative staining of bacteria, 2% uranyl acetate was used for contrast enhancement following pre-fixation with 2% glutaraldehyde (*Rachel et al., 2010*). Finally, transmission electron microscopy was performed either on a JEOL JEM 2100, operated at 120 kV in combination with a fast-scan 2k x 2k CCD camera F214 (TVIPS, Gauting, Germany) for negatively

stained samples or on a Zeiss EM 912 equipped with an integrated OMEGA energy filter and operated at 80 kV in the zero-loss mode for ultrathin sections.

## Phylogenetic analyses

We assembled a local databank of 205 complete genomes from a wide representative sampling of Firmicutes, including 38 Negativicutes and 7 Halanaerobiales genomes available at the beginning of this analysis (*Supplementary file 1*). Exhaustive HMM-based homology searches were carried out on this genome databank by using the HMMER package (*Johnson et al., 2010*) and as queries the HMM profiles of the complete set of 54 bacterial ribosomal proteins from the Pfam 29.0 database (http://pfam.xfam.org, *Finn et al., 2016*). Additional searches with tblastn (*Altschul et al., 1997*) were used to identify eventually misannotated homologues in some genomes. Because it is unclear which bacterial phylum is closest to the Firmicutes, we included as outgroup 13 taxa representatives of eight major bacterial phyla (2 Actinobacteria; 2 Cyanobacteria; 1 Deinococcus; 2 Proteobacteria; 1 Spirochaetes; 3 Flavobacteria/Bacteroidetes/Chlorobi; 2 Plactomyces/Chlamydia). Seven ribosomal proteins (S2, S4, S14, S21, L25, L30, L33) that were absent from >50% of the considered genomes or had paralogous copies making difficult the identification of orthologues were discarded from the analysis. The remaining 47 single protein data sets were aligned with MUSCLE v3.8.31 (*Edgar, 2004*) with default parameters, and unambiguously aligned positions were selected with BMGE 1.1 (*Criscuolo and Gribaldo, 2010*) and the BLOSUM30 substitution matrix.

Trimmed datasets were concatenated by allowing a maximum of 11 missing sequences per taxon into a large character supermatrix (218 taxa and 5551 amino acid characters). PhyloBayes v3.3b (*Lartillot et al., 2009*) was used to perform Bayesian analysis using the evolutionary model CAT+GTR+$\Gamma_4$. Two independent chains were run until convergence, assessed by evaluating the discrepancy of bipartition frequencies between independent runs. The first 25% of trees were discarded as burn-in and the posterior consensus was computed by selecting one tree out of every two . A Maximum likelihood (ML) tree was also calculated from the ribosomal protein concatenate with PhyML v3.0 (*Guindon et al., 2010*) and the evolutionary model LG+$\Gamma_4$ (*Le and Gascuel, 2008*) as suggested by the Akaike Information Criterion implemented in ProtTest 3 (*Darriba et al., 2011*). Branch supports were estimated by standard nonparametric bootstrap based on 100 replicates.

In order to assess whether the ribosomal proteins carried a congruent phylogenetic signal, we compared each of the 47 corresponding individual gene trees with the Bayesian ribosomal protein concatenate tree by using the recently proposed 'Internode Certainty' (IC) criterion, which measures the existence of statistically supported conflicting splits between trees (*Kobert et al., 2016*). ML phylogenetic trees of individual genes were inferred by IQ-TREE v1.3.12 (*Nguyen et al., 2015*) with evolutionary model selected by optimizing the Akaike information criterion. In order to minimize the negative impact on IC estimation of the large irresolution within most of the single gene trees caused by the small number of aligned characters, all branches displaying <70% bootstrap support were collapsed. IC values were then estimated by RAxML 8.2.8 (*Stamatakis, 2006*) and reported on the concatenate tree.

Significance of 12 alternative tree topologies was assessed by the approximately unbiased (AU) test (*Shimodaira, 2002*). Each alternative topology was obtained by moving specific nodes on the Bayesian concatenate ribosomal protein tree by using Seaview v4.6 (*Gouy et al., 2010*). For each tree topology, log-likelihood per character was estimated by PhyML v3.0 with the evolutionary model LG+$\Gamma_4$. In order to estimate the AU test p-values associated to each topology, the resulting data were processed with CONSEL v0.20 (*Shimodaira and Hasegawa, 2001*) with default parameters.

For the LPS core gene analysis, homologues were searched by using Pfam HMM profiles for LpxA, LpxB, LpxC, and LpxD. The same approach as the one described above for ribosomal proteins was used to assemble a 4-gene supermatrix of 898 unambiguously aligned amino acid characters, which was analysed by PhyloBayes with the evolutionary model CAT+GTR+$\Gamma_4$. Congruence among the four markers was assessed by the IC criterion as described above.

## Protein annotation

Given the small number of markers analyzed and their frequently limited conservation at the sequence level, we followed a semi-manual procedure for annotation based on a combination of

profile-based homology searches, protein domain identification, and genomic synteny. Profile-based homology searches of specific OM markers were performed by using the HMMER package (*Johnson et al., 2010*). Initial searches were conducted by HMM on the local Firmicute genome databank by using standard Pfam domain models corresponding to a given protein of interest. The top-scoring hits were used to build new HMM models and perform a novel search. This process was iterated until no new hits were found. The resulting homologues were aligned and manually inspected in order to confirm homology and to eliminate divergent, partial or non-homologous sequences. Additional searches with tblastn (*Altschul et al., 1997*) were used to identify eventually misannotated homologues in some genomes. Protein domains were inspected by querying the Conserved Domain Database (CDD) at NCBI (*Marchler-Bauer et al., 2015*). Genomic synteny was investigated using the interactive web-based visualization tool SyntTax (*Oberto, 2013*). Local genomic alignments were generated using EasyFig (*Sullivan et al., 2011*) with a BLAST cutoff E-value of 0.0001. For localization prediction, we used the PSORT v3.0 server (http://www.psort.org/psortb/, *Yu et al., 2010*) with default settings for Gram-negative Bacteria. Protein families were assigned to Clusters of Orthologous Groups by searching the COG database, which was downloaded from the NCBI FTP server (ftp://ftp.ncbi.nih.gov/pub/wolf/COGs/) (*Tatusov et al., 2000*).

## Note

While this manuscript was in the last phase of revision, Tocheva et al. published a Perspective paper (Tocheva, EI, Ortega, DR & G Jensen. 2016. Sporulation, bacterial cell envelopes and the origin of life. *Nature Reviews Microbiology* 14, 533-542. doi:10.1038/nrmicro.2016.85). It extends the discussion of their previous hypothesis (*Tocheva et al., 2011*) by focusing on the origin of the outer membrane, and prompts for further genomic and evolutionary analysis, which is timely addressed in the present work.

## Acknowledgements

LA was the recipient of a postdoctoral fellowship from the Investissement d'Avenir Grant "Ancestrome" (ANR-10-BINF-01-01). DP is a scholar in the Pasteur-Paris University International PhD program. AK wishes to thank Uwe-G Maier for allocation of the EM facility in Marburg and Marion Debus for technical assistance. We thank the Collection of Bacteria of the Institut Pasteur for kindly providing the two strains used in this work for electron microscopy, and the Centre Informatique pour la Biologie (CIB, pole cluster) for providing computing facility. We also wish to thank Patrick Forterre for discussion during the preparation of this work.

# Additional information

### Funding

| Funder | Grant reference number | Author |
| --- | --- | --- |
| Agence Nationale de la Recherche | ANR-10-BINF-01-01 | Céline Brochier-Armanet Simonetta Gribaldo |

The funders had no role in study design, data collection and interpretation, or the decision to submit the work for publication.

### Author contributions

LCSA, DP, AK, AC, Acquisition of data, Analysis and interpretation of data, Drafting or revising the article; BD, CB-A, CB, Analysis and interpretation of data, Drafting or revising the article; SG, Conception and design, Acquisition of data, Analysis and interpretation of data, Drafting or revising the article

### Author ORCIDs

Simonetta Gribaldo, http://orcid.org/0000-0002-7662-021X

# Additional files

## Supplementary files

• Supplementary file 1. Full distribution and accession numbers of the protein families discussed in the text.

• Supplementary file 2. Full distribution and accession numbers of the four core LPS genes in a local databank of 121 complete genomes representatives of all major bacterial phyla as discussed in the text. NA indicates that the taxon is unlisted in the NCBI Taxonomy database, while NH indicate that no homologues were found based on HMM search.

## Major datasets

The following dataset was generated:

| Author(s) | Year | Dataset title | Dataset URL | Database, license, and accessibility information |
|---|---|---|---|---|
| Luisa CS Antunes, Daniel Poppleton, Andreas Klingl, Alexis Criscuolo, Bruno Dupuy, Céline Brochier-Armanet, Christophe Beloin, Simonetta Gribaldo | 2016 | Data from: Phylogenomic analysis supports the ancestral presence of LPS-outer membranes in the Firmicutes | http://dx.doi.org/10.5061/dryad.48m7c | Available at Dryad Digital Repository under a CC0 Public Domain Dedication |

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
