## [Decision Letter]

Thank you for submitting your article "Phylogenomic analysis supports the ancestral presence of LPS-outer membranes in the Firmicutes" for consideration by *eLife*. Your article has been reviewed by three peer reviewers, one of whom, Jonathan Eisen, served as guest Reviewing Editor, and the evaluation has been overseen by Richard Losick as the Senior Editor. The following individuals involved in review of your submission have agreed to reveal their identity: Andrew Roger (peer reviewer).

The reviewers have discussed the reviews with one another and the Reviewing Editor has drafted this decision to help you prepare a revised submission.

Summary:

The consensus is that this is an interesting topic and that the work has some novel findings. There is not complete agreement on the degree of novelty of the findings. One factor for which there was consensus is that the paper does not do a thorough enough job of discussing the differences in this paper compared to previous papers on the same general topic. Thus if this paper is revised we recommend that the authors make a more thorough comparison to prior work including a comparison of phylogenetic results, statistical support for results, and inferences about the evolution of diderms and monoderms. This should include a discussion of the papers referenced in the reviews (e.g., Jensen, Tocheva et al. 2011, Errington 2013, Vollmer 2011, Yutin and Galperin 2013) and others to cover the history of theories on this topic and also a paper just published recently "Evolutionary relationships of completely sequenced Clostridia species and close relatives" see http://www.ncbi.nlm.nih.gov/pubmed/2641069. See the individual reviews for more details.

Essential revisions:

In addition to the topic of the novelty of the work some other concerns were raised by the reviewers which would need to be addressed in any revision. These include

1) Phylogenetic analysis should be done of individual genes to examine congruence among different markers and not just a concatenation based phylogeny. See review #2 for more detail.

2) Release of alignments. The alignments used in this paper and the treefiles for the trees need to be released to a public site such as Figshare or Dryad.

3) Statistical analysis for phylogeny in Figure 2. See Review 2 for details but more information is needed on the statistical analysis of branching patterns in this tree and additional statistical analyses are recommended.

4) Figure quality and amount. Some concern was expressed about the number of figures and the difficulty in reading some of them. It is recommended that the authors consider moving some Figures to supplemental material and redoing some figures to make them more readable.

5) Presence of second membrane in some species listed as monoderms. See Review 3 for details but this issue should be addressed and discussed in any revision.

6) Figure 1. There was some concern that the structures pointed to by the authors may be the result of artifacts. This issue (either further justification of this method or use of an alternative method) should be addressed in any revision.

7) Incomplete description of methods. See Review 1 in particular for more details but many of the methods used in the paper are not described in enough detail for readers to know what was done or for anyone to reproduce the work. A full description of all methods used needs to be included in any revision.

8) The computational workflows used in the paper (e.g., for identifying homologs and for generating and trimming alignments) are not tested in any obvious way. Some further discussion of the workflows they have used and any evidence regarding how well they work (e.g., alignment quality) would be helpful.

9) Support for results and conclusions is overstated in the text. See Review 1 in particular but there are many parts of the paper where the wording used is too strong in how it implies statistical or other support for their conclusions. In addition, many claims are made without referencing results or citing the literature. Any claims in the paper need to be supported either by results in the paper or by referencing the literature.

*Reviewer #1:*

Overall, I believe this paper is novel and represents some important contributions to the field of evolutionary biology especially in regard to the evolution of major groups of bacteria and their cell envelopes. And the detailed analysis of the phylogeny and the genomes and predicted pathways is very useful and important. In general, I believe many of their conclusions are likely to be supported by the data. However, there are some significant limitations of the manuscript in its current form that prevent me from being able to go beyond "likely to be supported" to saying "are supported". These areas need addressing before I would recommend publishing it in *eLife*.

Below are the major areas I think need addressing:

1) Methods description is generally incomplete and inadequate: much more detail needed including genome IDs, how COGs generated, BLAST parameters, HMM parameters, how outgroups were chosen, use of synteny and other information to predict function, how various web tools were run, parameters for many programs, what types of computers used, and much, much more. The methods are in many cases simply insufficiently described to be able to evaluate some sections of the paper.

2) Limited assessment of how well their phylogenomic approach works. It is good to see some high PP and other scores for the trees but some more analysis showing the validity of their results is important. For example, how do the branching patterns the see compare to those seen in other studies (including within each group)? Do they have any data showing their identification of orthologs works well? What about their alignemtns and identification of poorly aligned regions?

3) All alignments used in different phylogenomic analyses and functional predictions need to be released publicly.

4) There is repeated use of overly strong terms to describe the results and conclusions. For example the term "indicate' or "strongly support" or their varieties are used extensively but in most cases are not appropriate (e.g., in regard to what their phylogenetic analysis implies or what the presence of genes in the genomes implies). The authors should go through the entire paper and rephrase many sections. In essence the problem is that the wording repeatedly over inflates the results. I agree in most cases that one could conclude "we believe this result supports this model" but unless they present statistics or explicit hypothesis testing that is as far as they should go in the wording.

5) Many statements are made in the text without either results or citations to back them up.

*Reviewer #2:*

In this manuscript the authors marshal several types of evidence (phylogenomic, TEM, and genome content analysis) to suggest that two very distinct types of bacteria – the Negativicutes and the Haloanaerobiales which belong to the single-membrane bounded ('monoderm') Firmicute clade of bacteria, (an important 'gram-positive' bacterial group) – actually possess outer membranes and are 'diderm'. Furthermore, the authors show that like classical gram-negative bacteria these taxa retain a number of the molecular machines that function with the outer membrane in 'diderm' bacteria. Their phylogenetic positions, as assessed by a 47 ribosomal protein super-matrix, indicate that they emerge from within two distinct clades in the Firmicutes suggesting that the 'monoderm' nature of other Firmicutes has evolved multiple times from 'diderm' firmicute ancestors. The authors go on to describe features of the outer membrane which must be ancestral to Firmicutes and lost separately in the monoderm members (flagellar genes, LPS genes and pilus genes). These are extremely interesting findings and are important contributions to the understanding of bacterial genome and cell evolution.

In general I think the study is quite thorough and the conclusions are very interesting – especially with regard to the polyphyly of 'monoderm' condition within Firmicutes. This should serve as a strong argument against the claims that loss of the outermembrane is extremely rare and occurred possibly only once in prokaryotic evolution (e.g. as repeatedly claimed by Cavalier-Smith over the last few decades).

Below I itemize some suggestions for revisions in the manuscript:

1) The phylogenetic analyses of the 47 ribosomal protein genes are well done in general. However, these days there is a lot of talk about lateral gene transfer in bacterial genome evolution and I think it is important to do some kind of analyses to show that the authors believe that all of these proteins have evolved 'vertically' – i.e. demonstrate that their single phylogenies are generally congruent with each other. There are a number of published methods to do this rigourously that are very involved (e.g. as reviewed by Leigh et al. (2011) Genome Biol. Evol. 3:571-87). However I think simpler methods could also be used. For example, the authors could report the proportion of genes in the concatenate that support each of the splits in the ML tree that they show. They could also check to see whether any of the single gene trees conflict with the concatenated tree with >70% bootstrap support for the conflicting split. Another measure that the authors could consider is 'internode certainty' (IC) that has recently been extended to the case where single gene trees do not contain all taxa of the full tree (i.e. deal with missing data) – see Kobert et al. (2016) Mol. Biol. Evol. (advanced access); this method has been implemented in RAxML and could be easily applied to the data set.

Note that this same criticism applies to the concatenation of the 7 components of the LPS biosynthesis pathway (Figure 3). Some attempt to evaluate congruence amongst these markers should be made.

The general point here is that there needs to be some evidence that the markers being used in these super-matrices are all congruent (or at least congruent for the relevant parts of the tree being discussed).

2) Figure 2 shows the Bayesian phylogeny estimated and posterior probabilities on splits. However, there are second numbers on those splits that I am guessing are bootstrap values. However, the figure legend doesn't mention what they are. If they are bootstrap values, then the fact that they are sometimes quite a bit lower than the posterior probability should be acknowledged in the discussion of the phylogenetic positions of the Halanaerobiales and Negativicutes. For example, the latter taxa maybe 'nested' within the Clostridia implying many independent losses of LPS as shown in Figure 3, yet all of the backbone branches in the tree seem to be supported by very weak bootstrap values (if this values are what they seem to be). This is highly relevant to the discussion and should be mentioned. Specifically, the authors should specify a range of independent numbers of losses of LPS for Figure 3. Furthermore, probably the authors should do topology tests (e.g. AU tests) to see if a 'basal' branching position of the Negativicutes within the Clostridia can be excluded statistically. I suspect not. It would also be useful to see bootstrap values on the LPS tree since posterior probabilities don't adequately capture the uncertainty in branching patterns.

3) There are far too many figures in the manuscript. Some are dispensable or should be moved to supplementary information. For example, Figure 6 is unreadable in the colours given and is not interesting (redo it without coloured background and put it in the supplementary material). Figures 7, 8 and 9 are interesting but large. Perhaps the authors can condense the information shown in these figures and put them as 3 panels of one figure. Alternatively pick two of them to make into two panels and put the third in supplementary information.

*Reviewer #3:*

The paper by Antunes et al. describes detailed phylogenomic analyses of two diderm classes, the Negativicutes and Halanaerobiales that belong to the classically monoderm phylum Firmicutes. The results indicate shared outer membrane features between the two families that are also present in other diderm phyla. The authors conclude that since the outer membrane features are ancient in all diderm phyla, the presence of monoderm phyla in the current tree of bacteria is due to loss of the outer membrane from a common diderm ancestor.

I find the analyses thorough and incorporating significant amount of genomic data. However, these analyses are very similar to already published data and support published conclusions. In particular, previous phylogenetic analyses by Grant Jensen of the lps synthesis pathway, Omp85, TolC, and the flagellar genes *flgH* and *flgI* have already shown "no evidence to suggest that recent lateral gene transfer played a role in its acquisition of an OM. The possibility of convergent evolution of the OM in the Veillonellaceae is ruled out by the presence of many unambiguous homologs of different subsystems associated with OM biogenesis and function in other Gram-negative phyla." Please refer to Figure S7 and the Discussion of Tocheva et al. 2011 Cell "Peptidoglycan remodeling and conversion of an inner membrane into an outer membrane during sporulation". A phylogenetic tree of the Omp85 analysis shows the relationship between all diderm phyla and the lack of relationship with any one particular diderm phylum indicating that the Omp85 in A. longum (member of Veillonellaceae/Negativicutes) is an ancient feature. This is exactly what Antunes et al. claim with their more extensive analysis and present the idea as novel. It is important to note that the following hypothesis was also put forward by Jensen and co-authors: "We are only proposing that a single membraned last common ancestor of both Firmicutes and Gram-negatives possessed some primitive sporulation/phagocytotic-like ability, which gave rise to the OM. From this double-membraned sporulation ancestor many phyla could have diverged. Any one of these could have lost the OM and given rise to the Firmicutes. Other phyla could have lost the ability to sporulate at different times, giving rise to the current models of microbial phylogeny." These findings were further discussed and supported by reviews and articles written by Errinton 2013 Open Biol "L-form bacteria, cell walls and the origins of life", Vollmer 2011 Nat Chem Biol "Bacterial outer membrane evolution via sporulation?" and Yutin and Galperin 2013 Evniron Microbiol "A genomic update on clostridial phylogeny: Gram-negative spores formers and other misplace clostridia."

A major point that the authors fail to address is the presence of a second membrane in certain species that lack LPS/OMPs. For example, they claim that Thermotoga and Actinobacteria are monoderm phyla, however, Thermotoga species clearly have a unique OM and so do Mycobacteria (members of Actinobacteria). The authors need to address the difference between diderm and outer membranes containing LPS/OMPs.

Overall, I find the paper by Antunes et al. to support already existing claims by elaborating on the analysis (Negativicutes and Halanaerobiales). I didn't find any significant new claims presented here and therefore I don't consider this work novel enough for publication in *eLife*. This work would be appropriate for a more specialized journal since it provides detailed analysis of Negativicutes and Halanaerobiales and the relationship between the two.

---

## [Author Response]

*Summary:*

*The consensus is that this is an interesting topic and that the work has some novel findings. There is not complete agreement on the degree of novelty of the findings. One factor for which there was consensus is that the paper does not do a thorough enough job of discussing the differences in this paper compared to previous papers on the same general topic. Thus if this paper is revised we recommend that the authors make a more thorough comparison to prior work including a comparison of phylogenetic results, statistical support for results, and inferences about the evolution of diderms and monoderms. This should include a discussion of the papers referenced in the reviews (e.g., Jensen, Tocheva et al. 2011, Errinton 2013, Vollmer 2011, Yutin and Galperin 2013) and others to cover the history of theories on this topic and also a paper just published recently "Evolutionary relationships of completely sequenced Clostridia species and close relatives" see http://www.ncbi.nlm.nih.gov/pubmed/2641069. See the individual reviews for more details.*

We wish to thank the three reviewers and the editor for evaluating our submission and providing suggestions on how to make our analysis stronger. We agree that some improvement was necessary and we have now carefully revised the text by taking into account all comments. They helped improving the manuscript a lot, and we believe that its novelty with respect to previous publications is now clearly highlighted, as well the robustness of our results.

Regarding the novelty of our contribution, we acknowledge that Tocheva et al. 2011 and others made important contributions to the field. We have expanded the Introduction section to present these previously published data, which helps greatly to highlight the novelty of our analysis. Previous analyses were much less complete than ours, not because of inherent flaws but largely because of lack of genomic data for Halanaerobiales and Negativicutes, and results could not be interpreted unambiguously. The analysis by Tocheva et al. was very important, but largely focused on microscopy (their expertise field) rather than phylogenomics, and it presented a single tree (that of Omp85). Most importantly, they did not include Halanaerobiales.

We believe that the existence of a second diderm lineage in the Firmicute and its combined analysis with Negativicutes is one of the key originalities of our contribution. Moreover, we use much more genomes than any previous study. We would like to mention three major points that differentiate our analysis with respect to previously published data: (i) we robustly resolve the placement of the two diderm lineages in the Firmicutes phylogeny; (ii) we present for the first time a well-resolved LPS concatenate tree including a large taxonomic sampling of bacterial diversity, and clearly show that LPS-outer membranes are even more widespread in bacteria than currently known; (iii) we carry out a first detailed annotation of OM systems in these two atypical firmicute lineages, which complements the evolutionary discussion and provides important information on the Firmicute OM, guiding future experimental characterization. For all these reasons, we believe that our analysis provides robust phylogenomic data to the issue and, as such, is likely to become a highly cited reference, and of interest to the large audience of *eLife*.

We notice that Tocheva et al. have just published a Perspective paper in Nature Rev Microbiol that basically enlarges the discussion of their 2011 paper. The existence of a second diderm lineage in the Firmicutes, the Halanaerobiales, is not mentioned, and the authors conclude by welcoming future new data. This underlines the current strong interest of the scientific community on the issue and makes our analysis all the more timely. We have included a note to acknowledge this review at the end of the manuscript.

*Essential revisions:*

*In addition to the topic of the novelty of the work some other concerns were raised by the reviewers which would need to be addressed in any revision. These include*

*1) Phylogenetic analysis should be done of individual genes to examine congruence among different markers and not just a concatenation based phylogeny. See review #2 for more detail.*

Concatenation of ribosomal proteins has become a standard in the literature, as it has been shown early on that these markers rarely transfer (see Brochier et al. 2002 for example). Nevertheless, we have carried out the congruence tests suggested by reviewer 2. We observed no major incongruence, therefore justifying concatenation. The same was done for LPS gene concatenation. See more detailed answer to reviewers.

*2) Release of alignments. The alignments used in this paper and the treefiles for the trees need to be released to a public site such as Figshare or Dryad.*

These have been deposited to Dryad.

*3) Statistical analysis for phylogeny in Figure 2. See Review 2 for details but more information is needed on the statistical analysis of branching patterns in this tree and additional statistical analyses are recommended.*

We have carried out statistical tests on a number of alternative topologies for the Firmicutes tree. See more detailed answer to reviewers.

*4) Figure quality and amount. Some concern was expressed about the number of figures and the difficulty in reading some of them. It is recommended that the authors consider moving some Figures to supplemental material and redoing some figures to make them more readable.*

Yes, we agree. One figure has been removed, three figures have been moved to the supplementary material, and one was added to the main text. See more detailed answer to reviewers.

*5) Presence of second membrane in some species listed as monoderms. See Review 3 for details but this issue should be addressed and discussed in any revision.*

The Halanaerobiales have been known for a long time for having an outer membrane. More importantly, they have all the classical OM markers, which are specifically absent in monoderms and a clear indication of the presence of an outer membrane. See more detailed answer to reviewers.

*6) Figure 1. There was some concern that the structures pointed to by the authors may be the result of artifacts. This issue (either further justification of this method or use of an alternative method) should be addressed in any revision.*

Co-author Andreas Klingl, who is responsible for the electron microscopy platform at the University of Munich and who carried out the microscopy analyses is adamant on the quality of the images, and on the relevance of high pressure freezing for studying bacterial cell envelopes, as much as cryoelectron microscopy is. We would like however to precise that in this paper our aim was not to focus specifically on detailed microscopy analysis but to compare images of an outer membrane in Halanaerobiales and Negativicutes by the same technique, which had been lacking in current literature, and represents a good basis to discuss the genomics results. See more detailed answer to reviewers.

*7) Incomplete description of methods. See Review 1 in particular for more details but many of the methods used in the paper are not described in enough detail for readers to know what was done or for anyone to reproduce the work. A full description of all methods used needs to be included in any revision.*

We have expanded the Methods section (see more detailed answer to reviewers). Given the small number of markers analyzed and the delicate issue of de novo annotation, we did not use any automatic bioinformatics pipeline, which would have been less precise, but went mostly manually. We now give enough details so that any reader will be able to reproduce our results.

*8) The computational workflows used in the paper (e.g., for identifying homologs and for generating and trimming alignments) are not tested in any obvious way. Some further discussion of the workflows they have used and any evidence regarding how well they work (e.g., alignment quality) would be helpful.*

See comments above and more detailed answer to reviewers.

*9) Support for results and conclusions is overstated in the text. See Review 1 in particular but there are many parts of the paper where the wording used is too strong in how it implies statistical or other support for their conclusions. In addition, many claims are made without referencing results or citing the literature. Any claims in the paper need to be supported either by results in the paper or by referencing the literature.*

We praise such editorial policy of *eLife* in a time where evolutionary scientists may seem to be encouraged to oversell their results to get to the big journals, with deleterious consequences for the credibility of the discipline. That was however not our aim, and in the revised version of the manuscript we are careful to back up any strong statement by a strong result, while introducing nuance otherwise.

*Reviewer #1:*

*Overall, I believe this paper is novel and represents some important contributions to the field of evolutionary biology especially in regard to the evolution of major groups of bacteria and their cell envelopes. And the detailed analysis of the phylogeny and the genomes and predicted pathways is very useful and important. In general, I believe many of their conclusions are likely to be supported by the data. However, there are some significant limitations of the manuscript in its current form that prevent me from being able to go beyond "likely to be supported" to saying "are supported". These areas need addressing before I would recommend publishing it in eLife.*

Thank you for your comment on the importance of our work. We used a methodology that is relatively standard in phylogenomic studies. However, we acknowledge that more details were necessary. As for results, they have now been strengthened by additional tests allowing us to increase our confidence in their interpretation.

*Below are the major areas I think need addressing:*

*1) Methods description is generally incomplete and inadequate: much more detail needed including genome IDs, how COGs generated, BLAST parameters, HMM parameters, how outgroups were chosen, use of synteny and other information to predict function, how various web tools were run, parameters for many programs, what types of computers used, and much, much more. The methods are in many cases simply insufficiently described to be able to evaluate some sections of the paper.*

Details on the genomes have now been included in a table provided as additional data. Both the text and the Materials and methods section have been expanded to provide more detail on the approaches. The small number of proteins analyzed and the fact that they are often clustered on the genomes helped a lot manual annotation. COG categories were assigned by blasting the COG database with a representative of each protein family. We used personal computers in the lab for small-scale analyses and the server of the Institut Pasteur to run Bayesian trees. It is unusual to provide such kind of information in the M&M, and we would prefer not doing it.

*2) Limited assessment of how well their phylogenomic approach works. It is good to see some high PP and other scores for the trees but some more analysis showing the validity of their results is important. For example, how do the branching patterns the see compare to those seen in other studies (including within each group)? Do they have any data showing their identification of orthologs works well? What about their alignemtns and identification of poorly aligned regions?*

We now discuss more in detail in both the Introduction and Results sections the similarities and differences between our results and those previously published, including the recent analysis by Kunisawa 2015. Of course, we cannot fully compare all internal branching among our analysis and previous ones as none of these analyses use identical taxonomic sampling, although they generally agree. Our major aim in this paper was to discuss the placement of Halanaerobiales and Negativicutes and as such we compare specifically these placements with respect to what was published previously.

Concerning orthologs identification, we have tested the congruence of markers before concatenation as suggested by reviewer#2. Description of alignments and trimming methods was already included in the M&M of the originally submitted manuscript.

*3) All alignments used in different phylogenomic analyses and functional predictions need to be released publicly.*

We are providing all the single gene datasets and the corresponding concatenates used for the ribosomal protein tree and the LPS tree. As for functional prediction, all information was already included in a large table S1 with all accession numbers of the markers discussed in the text for all Halanaerobiales and Negativicutes. This should be sufficient.

*4) There is repeated use of overly strong terms to describe the results and conclusions. For example the term "indicate' or "strongly support" or their varieties are used extensively but in most cases are not appropriate (e.g., in regard to what their phylogenetic analysis implies or what the presence of genes in the genomes implies). The authors should go through the entire paper and rephrase many sections. In essence the problem is that the wording repeatedly over inflates the results. I agree in most cases that one could conclude "we believe this result supports this model" but unless they present statistics or explicit hypothesis testing that is as far as they should go in the wording.*

The text has been largely modified, and we have introduced nuances when appropriate.

5) Many statements are made in the text without either results or citations to back them up.

The text has been extensively revised to take into account these comments.

*Reviewer #2:*

*In this manuscript the authors marshal several types of evidence (phylogenomic, TEM, and genome content analysis) to suggest that two very distinct types of bacteria – the Negativicutes and the Haloanaerobiales which belong to the single-membrane bounded ('monoderm') Firmicute clade of bacteria, (an important 'gram-positive' bacterial group) – actually possess outer membranes and are 'diderm'. Furthermore, the authors show that like classical gram-negative bacteria these taxa retain a number of the molecular machines that function with the outer membrane in 'diderm' bacteria. Their phylogenetic positions, as assessed by a 47 ribosomal protein super-matrix, indicate that they emerge from within two distinct clades in the Firmicutes suggesting that the 'monoderm' nature of other Firmicutes has evolved multiple times from 'diderm' firmicute ancestors. The authors go on to describe features of the outer membrane which must be ancestral to Firmicutes and lost separately in the monoderm members (flagellar genes, LPS genes and pilus genes). These are extremely interesting findings and are important contributions to the understanding of bacterial genome and cell evolution.*

*In general I think the study is quite thorough and the conclusions are very interesting – especially with regard to the polyphyly of 'monoderm' condition within Firmicutes. This should serve as a strong argument against the claims that loss of the outermembrane is extremely rare and occurred possibly only once in prokaryotic evolution (e.g. as repeatedly claimed by Cavalier-Smith over the last few decades).*

Thank you, this is a very pertinent comment. We now mention this point in the Discussion section.

*Below I itemize some suggestions for revisions in the manuscript:*

*1) The phylogenetic analyses of the 47 ribosomal protein genes are well done in general. However, these days there is a lot of talk about lateral gene transfer in bacterial genome evolution and I think it is important to do some kind of analyses to show that the authors believe that all of these proteins have evolved 'vertically' – i.e. demonstrate that their single phylogenies are generally congruent with each other. There are a number of published methods to do this rigourously that are very involved (e.g. as reviewed by Leigh et al. (2011) Genome Biol. Evol. 3:571-87). However I think simpler methods could also be used. For example, the authors could report the proportion of genes in the concatenate that support each of the splits in the ML tree that they show. They could also check to see whether any of the single gene trees conflict with the concatenated tree with >70% bootstrap support for the conflicting split. Another measure that the authors could consider is 'internode certainty' (IC) that has recently been extended to the case where single gene trees do not contain all taxa of the full tree (i.e. deal with missing data) – see Kobert et al. (2016) Mol. Biol. Evol. (advanced access); this method has been implemented in RAxML and could be easily applied to the data set.*

*Note that this same criticism applies to the concatenation of the 7 components of the LPS biosynthesis pathway (Figure 3). Some attempt to evaluate congruence amongst these markers should be made.*

*The general point here is that there needs to be some evidence that the markers being used in these super-matrices are all congruent (or at least congruent for the relevant parts of the tree being discussed).*

Concatenation of ribosomal proteins has become a standard in building species phylogenies, as they have been shown in the past to be less prone to horizontal gene transfer (see for example Brochier et al. 2002). To our knowledge, none of the previous analyses on the phylogeny of Firmicutes (and in general all recently published large-scale phylogenies) contain such individual gene congruence tests (see for example Yutin et al.; Kunisawa. etc.). However, we agree that such analyses can strengthen our results. It has to be noted, nevertheless, that no congruence test is perfect, in particular those based on strict topology comparison, and there is still much room for development of strategies and methodologies. Thank you for providing precise suggestions on possible software. We have used IC tests as they deal with different taxonomic sampling in the single gene datasets. Of course, ribosomal proteins being mostly small, the phylogenetic signal carried by each single marker is accordingly weak. As a result, most single gene trees were largely unresolved, and most topological differences among trees are likely due to stochastic errors rather than true incongruent signal. However, we found very few split conflicting with the concatenate tree at >70% BV, as you suggested (IC<1), and these do not concern any major high rank nodes, i.e. those leading to the monophyly of major classes and orders, which are important to our discussion. This analysis is now provided as Figure 2—figure supplement 1. All raw data are available for the reviewers in Dryad (folder IC-tests; [2]).

Same results apply for the LPS genes. We decided to be more conservative and make a new analysis including only four core LPS genes instead of the seven genes previously used, because they are conserved in all bacteria containing LPS-OM and in order to avoid too much missing data that might have caused artifacts in the concatenate tree. Moreover, these four genes display conserved synteny across large taxonomic distances, strongly suggesting that they carry a congruent signal. Indeed, the resulting tree is fully consistent with the previous one based on 11 genes, and moreover it recovers monophyly of a few additional phyla. Congruence among the four markers was confirmed by IC tests, which showed no strong conflicting splits concerning the monophyly of major bacterial phyla at >70% BV (now shown in Figure 4—figure supplement 1).

*2) Figure two shows the Bayesian phylogeny estimated and posterior probabilities on splits. However, there are second numbers on those splits that I am guessing are bootstrap values. However, the figure legend doesn't mention what they are. If they are bootstrap values, then the fact that they are sometimes quite a bit lower than the posterior probability should be acknowledged in the discussion of the phylogenetic positions of the Halanaerobiales and Negativicutes. For example, the latter taxa maybe 'nested' within the Clostridia implying many independent losses of LPS as shown in Figure 3, yet all of the backbone branches in the tree seem to be supported by very weak bootstrap values (if this values are what they seem to be). This is highly relevant to the discussion and should be mentioned. Specifically, the authors should specify a range of independent numbers of losses of LPS for Figure 3. Furthermore, probably the authors should do topology tests (e.g. AU tests) to see if a 'basal' branching position of the Negativicutes within the Clostridia can be excluded statistically. I suspect not. It would also be useful to see bootstrap values on the LPS tree since posterior probabilities don't adequately capture the uncertainty in branching patterns.*

Maximum likelihood analysis of the ribosomal protein concatenate with the site homogeneous model LG was basically consistent with the Bayesian tree but did not succeed in resolving the deepest nodes, in particular the four nodes you noticed with very low BV% (the ML tree is now provided as Figure 2—figure supplement 2). In fact, Bayesian analysis with the site heterogeneous model CAT+GTR frequently outperforms site homogeneous models such as LG that are implemented in maximum likelihood frameworks, and provides better resolution for deep, difficult nodes (Lartillot & Philippe 2004).

This is clearly evident when we look at the LPS concatenate analysis. While the Bayesian tree with CAT+GTR resolves the monophyly of basically all major bacterial phyla, the ML tree with LG is completely unresolved (the tree is given to the reviewers for consultation, but we do not believe necessary to show this tree or map these BV% onto the Bayesian tree, as they do not contribute any meaningful data to the discussion).

Concerning AU tests on alternative topologies of the ribosomal tree, these are run with a site homogeneous model, and are expected to reflect the same uncertainty as the ML tree. Unfortunately no hypothesis testing is currently available in a Bayesian framework, so we could not compare the two models on these alternative topologies. We nevertheless ran AU tests with the LG site homogeneous model on 12 alternative topologies where Negativicutes or Halanaerobiales were moved “up and down” the n nodes separating them (N_1_-N_6_ for the topologies involving moving the Negativicutes; H_1_-H_6_ for the topologies moving the Halanaerobiales, Figure 2—figure supplement 3 and supplementary data). These alternative topologies imply a range of losses from 2 to 5. The Bayesian topology (H_0_N_0_) was the preferred one (5 losses). Then, two topologies only could not be rejected by the data, where the Negativicutes branched a bit earlier in the Clostridia (N_1_ and N_2_, implying 4 and 3 losses respectively). Importantly, the alternative topologies presenting monophyly of Halanaerobiales and Negativicutes for these ribosomal markers was strongly rejected, as well as all the topologies where the Halanaerobiales were moved away from the root of the Firmicute tree. This is consistent with the separate origins of the two diderm lineages, and with the hypothesis of a diderm ancestor for the Firmicutes. Therefore, although this test with a less performing model than CAT+GTR, it adds robustness and completeness to our results and is now included in the revised version. This analysis is now provided as Figure 2—figure supplement 3.

*3) There are far too many figures in the manuscript. Some are dispensable or should be moved to supplementary information. For example, Figure 6 is unreadable in the colours given and is not interesting (redo it without coloured background and put it in the supplementary material). Figures 7, 8 and 9 are interesting but large. Perhaps the authors can condense the information shown in these figures and put them as 3 panels of one figure. Alternatively pick two of them to make into two panels and put the third in supplementary information.*

We agree. We removed Figure 6 and moved Figures 7, 8 and 9 to the supplementary material. They are now Figure 5—figure supplement 1, Figure 5—figure supplement 2 and Figure 5—figure supplement 3, respectively.

*Reviewer #3:*

*The paper by Antunes et al. describes detailed phylogenomic analyses of two diderm classes, the Negativicutes and Halanaerobiales that belong to the classically monoderm phylum Firmicutes. The results indicate shared outer membrane features between the two families that are also present in other diderm phyla. The authors conclude that since the outer membrane features are ancient in all diderm phyla, the presence of monoderm phyla in the current tree of bacteria is due to loss of the outer membrane from a common diderm ancestor.*

*I find the analyses thorough and incorporating significant amount of genomic data. However, these analyses are very similar to already published data and support published conclusions. In particular, previous phylogenetic analyses by Grant Jensen of the lps synthesis pathway, Omp85, TolC, and the flagellar genes flgH and flgI have already shown "no evidence to suggest that recent lateral gene transfer played a role in its acquisition of an OM. The possibility of convergent evolution of the OM in the Veillonellaceae is ruled out by the presence of many unambiguous homologs of different subsystems associated with OM biogenesis and function in other Gram-negative phyla." Please refer to Figure S7 and the Discussion of Tocheva et al. 2011 Cell "Peptidoglycan remodeling and conversion of an inner membrane into an outer membrane during sporulation". A phylogenetic tree of the Omp85 analysis shows the relationship between all diderm phyla and the lack of relationship with any one particular diderm phylum indicating that the Omp85 in A. longum (member of Veillonellaceae/Negativicutes) is an ancient feature. This is exactly what Antunes et al. claim with their more extensive analysis and present the idea as novel. It is important to note that the following hypothesis was also put forward by Jensen and co-authors: "We are only proposing that a single membraned last common ancestor of both Firmicutes and Gram-negatives possessed some primitive sporulation/phagocytotic-like ability, which gave rise to the OM. From this double-membraned sporulation ancestor many phyla could have diverged. Any one of these could have lost the OM and given rise to the Firmicutes. Other phyla could have lost the ability to sporulate at different times, giving rise to the current models of microbial phylogeny." These findings were further discussed and supported by reviews and articles written by Errinton 2013 Open Biol "L-form bacteria, cell walls and the origins of life", Vollmer 2011 Nat Chem Biol "Bacterial outer membrane evolution via sporulation?" and Yutin and Galperin 2013 Evniron Microbiol "A genomic update on clostridial phylogeny: Gram-negative spores formers and other misplace clostridia."*

We appreciate the paper by Jensen and colleagues and we are sorry if we gave the impression that we did not acknowledge it sufficiently. According to the comments of the other two reviewers, we now give more detail in both the introduction and Results section on the analyses presented by Tocheva et al., as well as the other papers that dealt with the issue. However, we gently disagree with the claim that our analyses are very similar to the ones presented in that paper and do not provide any novel significant contribution. We present a lot of novel data that complement and extend previous studies including that of Tocheva et al. and provide much more robust results. To our knowledge, the only tree of an OM marker that was shown in Tocheva et al. was that of Omp85, while the other markers were only mentioned but not shown, so it is difficult to compare our results to theirs. In order to facilitate discussion and comparison with our results, we include in this response the Supplementary Figure S7 of Tocheva et al. that is mentioned above, for your convenience.

More importantly, Tocheva et al. did not include Halanaerobiales in their analysis, which is the major key that brings novelty to our study.

*A major point that the authors fail to address is the presence of a second membrane in certain species that lack LPS/OMPs. For example, they claim that Thermotoga and Actinobacteria are monoderm phyla, however, Thermotoga species clearly have a unique OM and so do Mycobacteria (members of Actinobacteria). The authors need to address the difference between diderm and outer membranes containing LPS/OMPs.*

We might have been unclear in the text. We only say that these phyla do not have outer membranes with LPS, not that they are monoderms. We have now clarified this point in the revised version. The fact that the large majority of bacterial phyla have some sort of outer membrane is now clearly stated and strengthens the hypothesis of the monoderm phenotype being a derived feature.

*Overall, I find the paper by Antunes et al. to support already existing claims by elaborating on the analysis (Negativicutes and Halanaerobiales). I didn't find any significant new claims presented here and therefore I don't consider this work novel enough for publication in eLife. This work would be appropriate for a more specialized journal since it provides detailed analysis of Negativicutes and Halanaerobiales and the relationship between the two.*

As also noted by the other two reviewers, we provide indeed a large number of novel results (even more in the revised version). The inclusion of Halanaerobiales as a second diderm lineage in the Firmicutes and its comparison to the Negativicutes is not an irrelevant issue of interest to a specialized audience, but is rather key to the discussion. We provide robust support for an existing claim that was based on much weaker and partial genomic data and met no consensus in the literature. Our analysis nicely complements and extends previous studies, including the one by Tocheva et al.